# Identification of Cement Pavement with Temperature Effect and Evaluation of Polymer Grouting Effect

**DOI:** 10.3390/polym15092207

**Published:** 2023-05-06

**Authors:** Xifeng Du, Haoyuan Cheng, Shengjie Xu, Wenjun Pei

**Affiliations:** School of Water Conservancy and Civil Engineering, Zhengzhou University, 100 Science Rd., Zhengzhou 450001, China; haoyuan@gs.zzu.edu.cn (H.C.); 202011221010275@gs.zzu.edu.cn (S.X.); p13592157252@163.com (W.P.)

**Keywords:** FWD, temperature, polymer grouting, stripping identification

## Abstract

The falling weight deflectometer (FWD) detection system benefits from its outstanding characteristics of no damage, fast speed, and high precision. The warping deformation of cement concrete pavement occurs due to the temperature difference along the depth of the slab, which makes FWD detect different results under different temperature fields. In this study, we systematically carried out the cement pavement’s temperature field and deflection test. The experimental data were analyzed to obtain the temperature variation law of the top and bottom of the pavement slab every day. By establishing a three-dimensional finite element model of cement pavement with a multi-layer elastic foundation type, the influence of the temperature difference at the bottom of the slab on the deflection of the center point of the slab corner load under different working conditions, different seasons, different loads and whether there is polymer filling in the void area was studied. We summarize the correlation between the temperature difference and the influence coefficient and propose the cement pavement void identification and polymer grouting effect evaluation method considering the temperature effect.

## 1. Introduction

With the rapid development of road infrastructure, the demand for pavement nondestructive testing equipment and related testing technology has become increasingly urgent. When using the traditional Beckman beam (BB) to detect the deflection value, subjective human factors and the indoor and outdoor experimental conditions during the test will significantly impact the test results [1]. The stability and reliability of the drop hammer deflectometer (now referred to as FWD) are reflected in its extensive use in regulations and various technical requirements [2]. Scholars have developed many determination methods for cement concrete pavement slab unloading based on FWD. Among them, the three-point regression method has outstanding advantages and is widely used in actual road void detection projects, which can solve the above problems well [3].

Different ambient temperature fields have different influences on the thermal expansion of cement concrete materials. Changing only the temperature field of the cement concrete pavement slab will cause various degrees of thermal expansion of concrete materials and influence the bending value detected by FWD. Through the analysis of a large amount of pavement temperature field monitoring data in the existing literature, we can find that the temperature field of cement concrete pavement is greatly affected by atmospheric temperature, solar radiation, and atmospheric precipitation. The heat conduction in pavement and heat exchange with the surrounding environment is complicated due to numerous influencing factors [4]. As a result, the bottom of the cement concrete pavement slab roof will produce a noticeable temperature difference, and the internal temperature difference will change periodically with time [5]. The different temperature boundaries of the top and bottom of the cement concrete pavement slab cause a nonlinear distribution of the temperature difference along the road panel in the vertical direction, which causes different degrees of temperature warping of the pavement slab. The temperature warping stress generated will then change the stress field of the pavement slab and ultimately affect the accuracy of FWD test results to determine the actual pavement condition [6]. The current research results ignore some influential factors or reduce the influence weight of many significant factors. However, there is still a gap in the actual situation, so the research on the pavement temperature field assessment method still needs to be improved. The focus of scholars and this paper’s core research point is how to accurately detect and effectively repair the concrete pavement and evaluate the repair effect.

Different ambient temperature fields have different influences on the thermal expansion of cement concrete materials. Changing only the temperature field of the cement concrete pavement slab will cause various degrees of thermal expansion of concrete materials and influence the bending value detected by FWD. Therefore, we studied the relationship between temperature difference and the influence of the temperature difference on the effectiveness of FWD test results, corrected the measured deflection values accordingly, and obtained the deflection values of all the test results at different temperature fields under the standard temperature or representative temperature of the road surface. In this way, we can judge the structural performance of cement concrete pavement more accurately, and the correlation between the temperature difference and the influence coefficient under corresponding circumstances is summarized. We put forward the method of cement pavement stripping identification and polymer grouting effect evaluation considering the temperature effect, and experiments verified the accuracy of the proposed method.

## 2. Long-Term Experimental Study on Temperature Field of Cement Road Panel

### 2.1. Test of Temperature Field Variation of Cement Pavement

The study of relevant literature [7,8,9,10] on the temperature field of cement concrete pavement in China and abroad found that there still is a gap between the current research results and the actual situation. In order to better simulate the actual performance of a cement concrete pavement board, this study used standard design parameters of cement concrete pavement to design outdoor full-size tests, as shown in Figure 1 and Table 1.

In the paving link of the above cement concrete surface layer, TCA-TIH-type resistive temperature and humidity sensors were embedded in the angle of the panel along the road and in the depth direction of the panel at 0, 2, 4, 6, 8, 12, and 24 cm. The temperature inside the panel in these areas was monitored and recorded in real-time by a KH400 G-type data acquisition system. See Figure 2 and Figure 3.

### 2.2. Temperature Analysis of the Top and Bottom of Cement Road Panels

By sorting out the variation law of the temperature field of cement concrete pavement and testing a large amount of data collected by various temperature sensors, March, April, and May were taken as spring, June, July, and August as summer, September, October and November as autumn, and December, January and February as winter, and the temperature of the top and bottom of the slab was divided into four seasons for centralized analysis, as shown in Figure 4.

By analyzing Figure 4, the following conclusions can be drawn. (1) The variation law of the temperature of the top of the plate was as follows: the temperature of the top of the plate increased, and the change rate gradually decreased from large to small in 6~12 h in spring and autumn and 6~14 h in summer and winter, whereas the temperature of the top of the plate decreased from large to small in 12~6 h in spring and autumn, and the change rate of the top of the plate decreased from large to small in 14~6 h in summer and winter. (2) Regarding the variation rule of the bottom temperature, the bottom temperature increased slowly, and the change rate decreased gradually in 6~20 h in spring, 10~18 h in summer, 12~18 h in autumn, and 12~20 h in winter, whereas it slowly decreased in 20~6 h on the second day in spring, 18~10 h on the second day in summer, 18~12 h on the second day in autumn, and 20~12 h on the second day in winter, and the change rate gradually decreased. (3) In terms of the variation law of the temperature difference at the bottom of the roof, the variation trend of the temperature difference at the bottom of the roof in each season was consistent with that of the maximum temperature. The zero temperature difference appeared in 6~8 h in the morning and 18~20 h in the afternoon in spring, 8~10 h in the morning and 18~20 h in the afternoon in summer and autumn, and 8~10 h in the morning and 16~18 h in the afternoon in winter.

### 2.3. Correlation between the Top and Bottom Temperatures of Cement Road Panels and Their Temperature Differences

#### 2.3.1. Correlation between Top and Bottom Temperature–Temperature Differences Based on Each Season (Holistic Analysis)

According to the linear fitting formula in Figure 5, Figure 6, Figure 7, Figure 8, Figure 9, Figure 10, Figure 11 and Figure 12, it can be found that: (1) there is an undeniable linear relationship between the temperature difference of the top of the cement concrete pavement slab and the bottom temperature of the roof, and the slope and intercept of the linear fitting relation are different in different seasons due to the influence of atmospheric radiation, atmospheric temperature, and other natural factors. (2) There is no apparent correlation between the bottom temperature of the cement concrete pavement and the bottom temperature difference of the roof, so the correlation between the bottom temperature and the temperature difference of the cement concrete pavement is not considered.

To sum up, the correlation between the temperature difference between the slab top and roof bottom of the cement concrete pavement in different seasons can be obtained as follows:

(1) The temperature difference between the slab top and roof bottom of the cement concrete pavement in spring:(1)$$\nabla T=0.5039\cdot T_{T}-6.8963\left(R^{2}=0.8828\right)$$

(2) The temperature difference between the slab top and roof bottom of the cement concrete pavement in summer:(2)$$\nabla T=0.8761\cdot T_{T}-28.432\left(R^{2}=0.9161\right)$$

(3) The temperature difference between the slab top and roof bottom of the cement concrete pavement in autumn:(3)$$\nabla T=0.3261\cdot T_{T}-3.0845\left(R^{2}=0.9244\right)$$

(4) The temperature difference between the slab top and roof bottom of the cement concrete pavement in winter:(4)$$\nabla T=0.9609\cdot T_{T}-3.0845\left(R^{2}=0.9584\right)$$
where ∇T is the temperature difference at the bottom of the pavement slab roof, °C, and *T_T_* is the pavement roof temperature, °C. As seen from Figure 5, Figure 6, Figure 7, Figure 8, Figure 9, Figure 10, Figure 11 and Figure 12, the *T_T_* coefficients in Equations (1)–(4) differ due to the different average temperatures of the plate roof in each season.

#### 2.3.2. Correlation between Daily Mean Bottom Temperature and Temperature Difference Based on Each Season (Daily Mean Method)

By analyzing the variation law of the fitting curve in Figure 13, Figure 14, Figure 15, Figure 16, Figure 17, Figure 18, Figure 19 and Figure 20, it can be found that: (1) there is a very obvious linear relationship between the average daily temperature at the top of the pavement and the average temperature difference at the bottom of the roof at the corresponding time; (2) there is little correlation between the average daily floor temperature and the average daily roof the temperature difference in each season, so the correlation analysis between the average floor temperature and the temperature difference is not considered.

To sum up, the correlation equation between the average daily temperature of the cement concrete pavement in different seasons and the temperature difference of the roof bottom can be obtained as follows:

(1) The difference between the average daily temperature of the cement concrete pavement slab top and the bottom temperature of the slab roof in spring:(5)$$\nabla T^{\prime }=0.9901\cdot {T^{\prime }}_{T}-21.188\left(R^{2}=0.9692\right)$$

(2) The average daily temperature difference between the top and bottom of the cement concrete pavement slabs in summer:(6)$$\nabla T^{\prime }=0.9298\cdot {T^{\prime }}_{T}-30.069\left(R^{2}=0.9626\right)$$

(3) The average daily temperature difference between the top of the cement concrete pavement slab and the bottom of the roof in autumn:(7)$$\nabla T^{\prime }=1.0283\cdot {T^{\prime }}_{T}-18.625\left(R^{2}=0.9552\right)$$

(4) The average daily temperature difference between the top of the cement concrete pavement slab and the bottom of the roof in winter:(8)$$\nabla T^{\prime }=0.9782\cdot {T^{\prime }}_{T}-3.482\left(R^{2}=0.9696\right)$$
where $\nabla T^{\prime }$ is the difference of average daily temperature between the top and bottom of the pavement slab, °C, and *T*′_*T*_ is the average daily top temperature of the pavement board, °C.

#### 2.3.3. Accuracy Analysis of the Correlation between Top Temperature and Temperature Difference in Each Quarter

The measured temperature difference between the top and bottom of the cement concrete pavement slabs was compared with the panel temperature difference calculated by the above holistic analysis method and daily mean value method, as shown in Figure 21.

By comparing the measured linear distribution of the temperature difference between the slab top and roof bottom of the cement concrete pavement with the distribution curve calculated by the overall analysis method and daily mean value method in Figure 21, the following conclusions can be drawn:

(1) In terms of the overall trend, the measured distribution curve of the temperature difference between the roof temperature and the bottom temperature of the roof, the overall analysis method, and the daily mean method shows that there was a positive linear relationship between the temperature difference at the bottom of the roof and the top temperature.

(2) In terms of accuracy, in spring and autumn, the variation trend of the daily mean method was quite different from that of the overall analysis method and the measured distribution curve, among which the overall analysis method was more consistent with the actual situation.

## 3. Identification Method of Cement Pavement Stripping Considering Temperature Effect

### 3.1. Finite Element Model of Cement Pavement under the Action of Temperature and Load Coupling Stress Field

Due to the change in the environment, the different temperature differences in the cement concrete pavement slab cause the temperature warping of the pavement slab to amount to different degrees and change the supporting conditions of the pavement slab base [11]. Therefore, the bending result will be affected to different degrees during the FWD detection, resulting in the misjudgment of the performance of the pavement slab. The bending basin obtained by FWD detection of cement concrete pavement has a good relationship with parameters such as pavement bottom emptying, joint load transfer efficiency, base modulus, and foundation modulus [12].

According to the literature [13,14], there are three basic assumptions when establishing a multi-layer elastic structure model of cement concrete pavement: (1) the materials of each structural layer are homogeneous and are isotropic linear elastomers, which are in the elastic stage and represented by elastic modulus and Poisson’s ratio; (2) except for the panel, all layers are infinite elastic bodies in the horizontal direction, and the lowest layer is a homogeneous semi-infinite body; (3) the contact surface between layers is assumed to be thoroughly combined, and the displacements and stresses in each direction of the contact surface are continuous. The force and displacement transfer occur only when the contact surface is in contact with the unloading area. Because the size of the cement panel in the plane direction is much larger than the size in the vertical direction, the concrete panel has high strength and a large elastic modulus, so it is necessary to select an appropriate mechanical analysis model for concrete panel modeling.

(1)Selection of material parameters of each structural layer

In actual pavement, the plane size of the base, subbase, and soil foundation is infinite, which is difficult to achieve in finite element modeling. Through referring to the literature [15], it can be found that in previous simulations of a semi-infinite space body, the expanded size model with a truncated boundary was generally adopted. In order to better simulate the actual cement concrete pavement structure, when using the ABAQUS finite element software for simulation calculations, the model size should be set to meet the condition that the distance load of the model should be sufficiently small compared to the distance displacement [16]. The material parameters of the matching cement concrete pavement lattice structure layer were selected, as shown in Table 2.

(2)The setting of the void region for finite element simulation

This paper has two types of board bottom release: board angle release and board edge release. The plate corner emptying area is set as a thin cylindrical plate with the corner vertex as the center of the circle, and the plate edge emptying is set as a thin rectangular plate with a certain width and depth with the panel width as the long side. The thin cylindrical plate has four types of radius, namely 0.25, 0.5, 0.75, and 1 m, and four types of depth of 0.5, 1, 1.5, and 2 cm, which are combined into a total of 16 types of unloading conditions. The thin cuboid plate has four types of the width of 0.25, 0.5, 0.75, and 1 m, and four types of the depth of 0.5, 1, 1.5, and 2 cm, which are combined into a total of 16 types of emptying conditions. The board angle and board edge emptying settings are shown in Figure 22.

(3)Load and boundary conditions of finite element model

The finite element model of cement concrete pavement has two types of load: temperature stress and load. The temperature stress is added to the roof bottom of the cement concrete pavement slab by directly inputting the corresponding value in the load setting of the initial step. In finite element analysis, the setting of model boundary conditions dramatically influences the accuracy of simulation results. Therefore, in the finite element simulation of cement concrete in this paper, panel boundary conditions were set as follows in the initial analysis step:(9)$$Slab-x(\mathrm{road}\;\mathrm{board}\;x\;\mathrm{direction})\;U_{2}=U_{R_{2}}=U_{R_{3}}=0$$
(10)$$Slab-y(\mathrm{road}\;\mathrm{board}\;y\;\mathrm{direction})\;U_{1}=U_{R_{1}}=U_{R_{3}}=0$$
where *U*_1_, *U*_2_ are the translational displacement in model 1 and model 2 directions, respectively, and $U_{R_{1}},\;U_{R_{2}},\;U_{R_{3}}$ are the rotation angles centered on axes 1, 2, and 3, respectively.

In this model, the soil base is not set, and the elastic modulus of 50 MPa is directly set at the bottom of the base as the boundary condition.

(4)The setting of contact conditions and mesh division of the finite element model

This model adopts the technology of non-uniform grid division [15]. Considering that this paper mainly studies the bending change law of the center point of the plate angle load under the combined action of temperature load and bending load, the grid division density of the cement concrete pavement slab is relatively large (the mesh density of the mesh distribution is 0.05, and the mesh density of the plate angle load area is 0.025). The meshing density in the base part is small (mesh seed density is 0.2). The effect is shown in Figure 23 and Figure 24.

### 3.2. Influence of Coupling Effect of Temperature and Load Stress Field on Road Slab Angle Deflection

According to the variation of the cloud image in Figure 25, it can be found that when the pavement panel edge is released (release width 0.5 m, release depth 0.5 cm, load 50 KN), the panel angle within the range of the negative temperature difference gradually warps up as the temperature difference increases. When the temperature difference is zero, only the load affects the plate angle. When the temperature difference is positive because the base has better support to the road panel, the angle of the plate is only concave relative to part of the plate, but the overall displacement is upward. In the model, the deformation data generated only by temperature stress in different seasons when the plate edge is hollowed out were extracted and plotted, as shown in Figure 26.

According to the analysis of the variation trend of the shape variable in different unloading conditions in different seasons in Figure 26, it can be found that: (1) the influence of the temperature difference on the deformation of the center point of the plate angular load in different unloading conditions in the same season is very similar in magnitude; (2) in terms of variation trend, the central point shape variables of plate angle load all decrease with the increase of the temperature difference in the negative temperature difference and increase with the increase of the temperature difference in the positive temperature difference.

The deformation data of the corner load center point of the finite element model of cement concrete pavement under different temperature differences and slab edge voids were extracted and plotted, as shown in Figure 27, Figure 28 and Figure 29.

By comparing the distribution curves of the unloading conditions in Figure 27, Figure 28 and Figure 29, it can be concluded as follows: (1) In terms of the deformation size caused by the coupling effect of temperature stress and load stress field, the coupling deformation of temperature stress and load stress field in different seasons and with the same temperature difference can be ignored under the influence of the unloading depth, and the shape variable increases with the increase of the unloading radius. (2) In terms of the variation trend of the coupling deformation between temperature stress and load stress field, the deformation generated by the coupling effect of temperature stress and load stress field under all unloading conditions in the four seasons decreases with the increase of the temperature difference in the range of −25~−5 °C and increases with the increase of the temperature difference in the range of −5~25 °C.

The center point of the angular load of the plate is only subjected to temperature stress and deformation under the coupling of temperature and load stress field. Formula (11) is introduced to calculate the corresponding bending value of the plate’s center point’s angular load at different temperature differences.
(11)$$D_{Temperature\;effect}=L_{Temperature\;stress\;and\;load}-L_{only\;load}$$
where $D_{Temperature\;effect}$ is the bending value of the center point of the angular plate load considering the effect of the temperature stress field, μm; *L_only load_* considers only the central point deformation under load, unit, μm; and $L_{Temperature\;stress\;and\;load}$ is the central point deformation under temperature stress only, μm.

Formula (11) was used to calculate the bending value of the center point of the plate angle load in each emptying condition in the four seasons and draw the figure, as shown in Figure 30, Figure 31 and Figure 32.

By comparing the bending data affected by temperature in different seasons and the distribution curves of the unloading conditions in Figure 30, Figure 31 and Figure 32, the following conclusions can be drawn: (1) in the four seasons, the bending size of the same temperature difference considering the influence of temperature on the unloading depth is negligible, but increases with the increase of the unloading radius; (2) in the four seasons, the bending considering the influence of temperature in each emptying condition decreases with the increase of the temperature difference at the negative temperature difference, and increases with the increase of the temperature difference at the positive temperature difference.

### 3.3. Identification Method of Cement Pavement Stripping Considering Temperature Effect

This study introduces the concept of the degree of influence of the temperature difference on the bending value of the center point of the plate angular load—the influence coefficient, which is expressed as *φ*. The calculation method is as follows:(12)$$\varphi =D_{Temperature\;effect}/D_{Temperature\;effect\;is\;not\;considered}$$
where φ is the influence coefficient of the temperature difference on the deflection value of the center point of angular plate load; and $D_{Temperature\;effect\;is\;not\;considered}$ is without considering the temperature effect—a deflection value of the plate corner load center point without considering the effect of temperature stress field, μm.

The larger the influence coefficient is, the greater the influence of the corresponding temperature difference on the bending value of the center point of the plate angle load, and vice versa.

Formula (12) was used to calculate the temperature influence coefficient at the center point of the plate angle load in each emptying condition in the four seasons. The figure was drawn, as shown in Figure 33.

By comparing the temperature influence coefficient data in different seasons and the distribution curves of each emptying condition in Figure 34, Figure 35 and Figure 36, the following conclusions can be drawn: (1) in the four seasons, the temperature influence coefficient of the same temperature difference is not affected by the depth of emptying and increases with the increase of the emptying radius, but the variation range is small; (2) in the four seasons, the temperature influence coefficient of all empty-working conditions decreases with the increase of the temperature difference at the negative temperature difference, and increases with the increase of the temperature difference at the positive temperature difference.

The relative errors of temperature influence coefficients in different emptying conditions in the four seasons under a three-level load relative to the mean value of temperature influence coefficients with the same temperature difference were calculated, and the relative error distribution was analyzed. It was found that the change rule of temperature influence coefficients in different emptying conditions in the four seasons under a three-level load could be approximately replaced by the change rule of their mean value, and the relative errors were all less than 5%. The average temperature influence coefficients of different emptying conditions in the four seasons under a three-level load were plotted, as shown in Figure 36.

The analytical formula for fitting the relationship between the temperature difference and influence coefficient is shown as follows:

When the temperature difference is negative:(13)$$\varphi =3\cdot E^{-4}\cdot \frac{\nabla T^{3}}{\nabla T_{0}^{3}}-2.14\cdot E^{-2}\cdot \frac{\nabla T^{2}}{\nabla T_{0}^{2}}-4.854\cdot E^{-1}\cdot \frac{\nabla T}{\nabla T_{0}}+1.006\;\left(RMSE=0.00167,MAE=0.003243\right)$$

When the temperature difference is positive:(14)$$\varphi =2\cdot E^{-5}\cdot \frac{\nabla T^{3}}{\nabla {T_{0}^{\prime }}^{3}}-1.1\cdot E^{-3}\cdot \frac{\nabla T^{2}}{\nabla {T_{0}^{\prime }}^{2}}+2.18\cdot E^{-2}\cdot \frac{\nabla T}{\nabla T_{0}^{\prime }}+1.0078\;\left(RMSE=0.00167,MAE=0.001303\right)$$
where ∇*T*_0_ is a panel plate bottom temperature difference of −1 °C; $\nabla T_{0}^{\prime \prime }$ is a panel plate bottom temperature difference of 1 °C; RMSE is the root mean square error; and MAE is the mean absolute error.

## 4. Evaluation of Polymer Grouting Effect Considering Temperature Effect

### 4.1. Analysis of the Change of Angle Bending of the Road Panel after the Side Cavity Grouting

The deformation data generated by the coupling effect of temperature stress and load (50 KN) stress field after plate edge caving grouting in different seasons were extracted and plotted, as shown in Figure 37, Figure 38 and Figure 39.

By comparing and analyzing the deformation data generated by the coupling effect of temperature stress and load stress field after the plate angle unloading grouting in different seasons and the distribution curves of each unloading condition in Figure 37, Figure 38 and Figure 39, it can be concluded as follows: (1) in the four seasons, the deformation generated by the coupling effect of temperature stress and load stress field with the same temperature difference was less affected by the depth and width of the unloading; (2) in the four seasons, the deformation caused by the coupling effect of temperature stress and load stress field under each emptying condition decreased with the increase of the temperature difference at the negative temperature difference, and increased with the increase of the temperature difference at the positive temperature difference.

By comparing and analyzing the bending data of the slab angle unloading grouting in different seasons with the influence of temperature and the distribution curves of each unloading condition in Figure 40, Figure 41 and Figure 42, the following conclusions can be drawn: (1) in the four seasons, the bending size of the slab angle unloading grouting in the same temperature difference with the influence of temperature can be ignored under the influence of the unloading depth and increases with the increase of the unloading width, but the variation range is small; (2) in the four seasons, the deflection considering the influence of temperature decreases with the increase of the temperature difference in the negative temperature difference, and increases with the increase of the temperature difference in the positive temperature difference.

### 4.2. Correlation Analysis between Panel Temperature Difference and Influence Coefficient after Side Cavity Grouting

Formula (12) was used to calculate the temperature influence coefficient at the center point of the plate angle load in each emptying condition in the four seasons and draw the figure, as shown in Figure 43, Figure 44 and Figure 45.

By comparing the temperature influence coefficient data in different seasons and the distribution curves of each emptying condition in Figure 43, Figure 44 and Figure 45, the following conclusions can be drawn: (1) in the four seasons, the temperature influence coefficient of the same temperature difference is not affected by the depth of emptying and increases with the increase of the emptying width, but the magnitude of the change is small; (2) in the four seasons, the temperature influence coefficient of all empty-working conditions decreases with the increase of the temperature difference at the negative temperature difference, and increases with the increase of the temperature difference at the positive temperature difference.

The relative error of the temperature influence coefficient under a three-level load, in the four seasons, under different emptying conditions, and after grouting concerning the average temperature influence coefficient of the same temperature difference was calculated. The above-calculated average temperature influence coefficient under different emptying conditions in the four seasons under a three-level load was sorted out and plotted, as shown in Figure 46.

The fitting analytic expressions within the range of negative and positive temperature differences are listed, and the relation between temperature difference and influence coefficient is sorted out as follows.

When the temperature difference is negative:(15)$$\varphi =4\cdot E^{-4}\cdot \frac{\nabla T^{3}}{\nabla T_{0}^{3}}-2.37\cdot E^{-2}\cdot \frac{\nabla T^{2}}{\nabla T_{0}^{2}}+5.216\cdot E^{-1}\cdot \frac{\nabla T}{\nabla T_{0}}+0.9824\;\left(RMSE=0.00167,MAE=0.004196\right)$$


When the temperature difference is positive:(16)$$\varphi =1\cdot E^{-4}\cdot \frac{\nabla T^{3}}{\nabla {T_{0}^{\prime }}^{3}}-5.4\cdot E^{-3}\cdot \frac{\nabla T^{2}}{\nabla {T_{0}^{\prime }}^{2}}+9.29\cdot E^{-2}\cdot \frac{\nabla T}{\nabla T_{0}^{\prime }}+0.9594\;\left(RMSE=0.00167,MAE=0.008759\right)$$


### 4.3. Experimental Verification

Based on the above research and analysis, the analytical formula of the temperature difference and temperature influence coefficient after grouting was obtained, and the evaluation method of the polymer grouting effect for cement concrete pavement considering the temperature benefit is summarized as follows:

1. An infrared temperature detector or other road temperature measuring equipment is used to detect the *i* pavement panel and obtain the road surface temperature as $t_{i}$;

2. After determining which season is the detection time, the corresponding calculation formula of the temperature difference between the top of the slab and the bottom of the slab roof is used to obtain the corresponding temperature difference $\nabla T_{i}$ of the bottom of the slab roof;

3. The temperature difference obtained in the previous step is calculated using the corresponding temperature difference–temperature influence coefficient formula, and the temperature influence coefficient $\varphi_{i}$ at the corresponding detection time is obtained;

4. At the same time as the above temperature detection, the edge of the load plate of FWD is positioned 25 cm away from the corner edge of the road slab, and the third-grade load bending detection is carried out. The bending point of the load center under the third-grade load is $w_{1},w_{2},w_{3}$;

5. The $w_{i1},w_{i2},w_{i3}$ obtained in the previous step are divided by the temperature influence coefficient obtained in the third step to calculate the board angle deflection values $w_{i1}^{\prime },w_{i2}^{\prime },w_{i3}^{\prime }$ without temperature influence. The three-point regression method is used to determine the pavement slab grouting repair without temperature influence.

We designed the correlation test between pavement temperature difference and temperature influence coefficient as follows: (1) we buried the temperature sensor; (2) we set the board side unloading. We then lifted the cement concrete pavement board without unloading in good condition using a crane. We used a sander to set the unloading area on the top surface of the base in the way shown in the Figure 47, Figure 48, Figure 49, Figure 50, Figure 51 and Figure 52, with 0.5 m as the unloading radius and 2 cm as the unloading depth; (3) the bending data in the unloading condition were collected by using a digital display to collect the bottom temperature data detected by the temperature sensor on the pavement board, using an infrared temperature detector to collect the top temperature of the pavement board and using a drop hammer bending instrument to collect the bending data at the center point of the cornering load of the board. The above data collection work time was once an hour; (4) after collecting the appropriate amount of data in the first three steps, the grouting hole layout was designed according to the technical requirements of polymer grouting, and the hollow panel is drilled, grouted and blocked; (5) the specific operation of data collection of bending and sinking after grouting of the hollow panel was the same as in step 3.

Through sorting out the above test data, the bending value corresponding to zero temperature difference in daily detection work was taken as the standard value of all bending and subsidence detection on that day. The ratio of all bending and subsidence detection values (including the bending and subsidence values in the case of zero temperature difference) to the above standard value was calculated, namely the value of the temperature influence coefficient, which was sorted out and drawn as the Figure 53, Figure 54, Figure 55 and Figure 56 below.

The bending value considering the influence of temperature was obtained by analyzing the change law of deformation only affected by temperature and deformation coupled with the load at the center point of the angular plate load and calculating the difference between the two shape variables. The change law of the above bending value under different emptying conditions and the temperature difference at the bottom of the plate roof was studied, and the conclusions were drawn as follows:

1. When the pavement is in a negative temperature difference, the angle of the pavement is warped upward relative to the center of the pavement. With the increase in the temperature difference, the deviation value will gradually decrease. When the pavement is in a positive temperature difference, the angle of the pavement is concave downward relative to the center of the pavement. The detection results will be more significant to a certain extent with the increase of the temperature difference, but the increase is slight, and then there is a downward trend. The cavitation depth does not affect the temperature influence coefficient of the same temperature difference. It increases with the cavitation’s width, but the change’s magnitude is small. The temperature influence coefficient of all empty-working conditions decreases with the increase of the temperature difference at the negative temperature difference. It increases with the increase of the temperature difference at the positive temperature difference. 2. In the case of edge emptying, when the edge emptying width is constant, the emptying depth has little influence on the correlation between the temperature difference and the temperature influence coefficient; when the venting depth is constant, the venting width has a specific effect on the correlation between the temperature difference and the temperature influence coefficient. It is verified that the temperature influence coefficient changes. This further shows that for considering the temperature effect, the cement pavement stripping identification and polymer grouting effect evaluation method is feasible.

## 5. Conclusions

Based on the actual temperature detection of cement concrete pavement and the three-dimensional finite element model of cement pavement of the multi-layer elastic foundation type, this paper proposes an identification method of cement pavement unloading and an evaluation method of polymer grouting considering the temperature effect. The main contents and conclusions are as follows:

1. Through the reanalysis of the temperature field test data of the cement concrete pavement, it was obtained that (1) the temperature of the top of the slab in each quarter presents a half-sinusoidal change with time, and the temperature of the bottom of the slab is relatively stable; (2) the variation range of the temperature at the top of the pavement is generally consistent with the temperature difference at the bottom of the roof.

2. By establishing a three-dimensional finite element model of cement pavement with a multi-layer elastic foundation type, the temperature difference at the bottom of the slab roof was studied under different working conditions, different loads, and with or without polymers when the slab edge is emptied. We propose the influence of the identification of cement pavement with temperature effect and the filling of polymer grouting effect evaluation method on the center point bending of the slab angle load. The correlation between the temperature difference and influence coefficient is summarized.

3. We analyzed the change of the angle deflection value of the road panel after the side unloading grouting and put forward the identification method of cement pavement unloading and the evaluation method of the polymer grouting effect considering the temperature effect. We verified the variation rule of the influence coefficient of the temperature field through a field test.

## Figures and Tables

**Figure 1 polymers-15-02207-f001:**
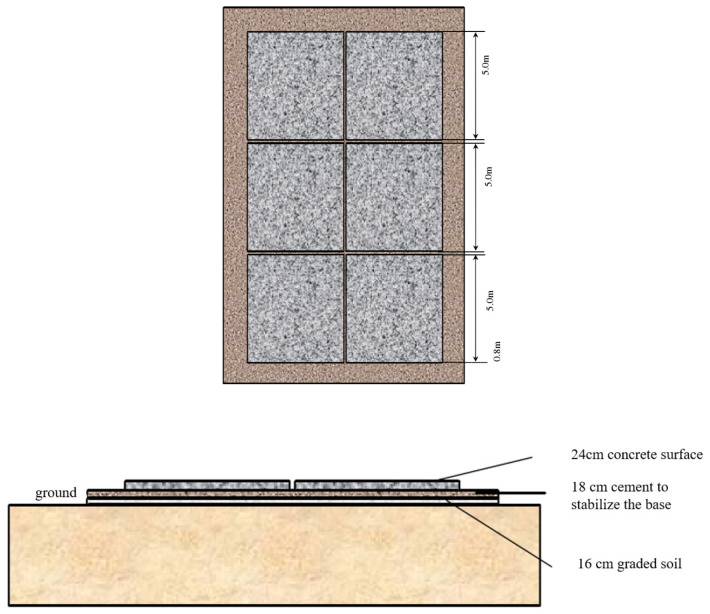
Schematic diagram of cement concrete pavement lattice structure layer.

**Figure 2 polymers-15-02207-f002:**
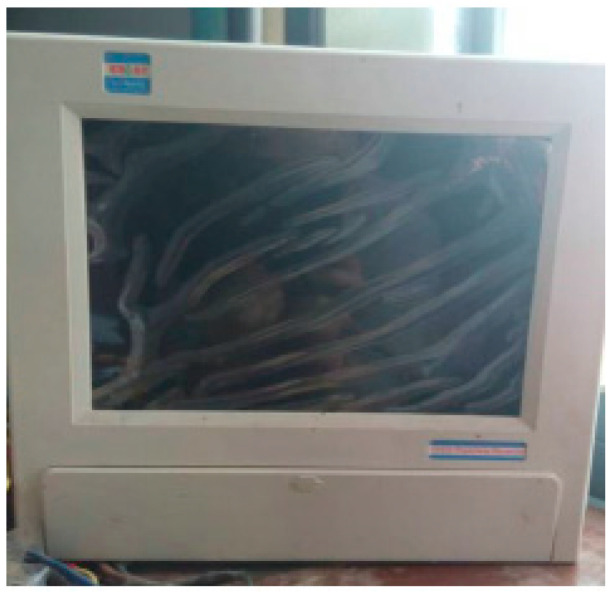
KH400G automatic data acquisition system.

**Figure 3 polymers-15-02207-f003:**
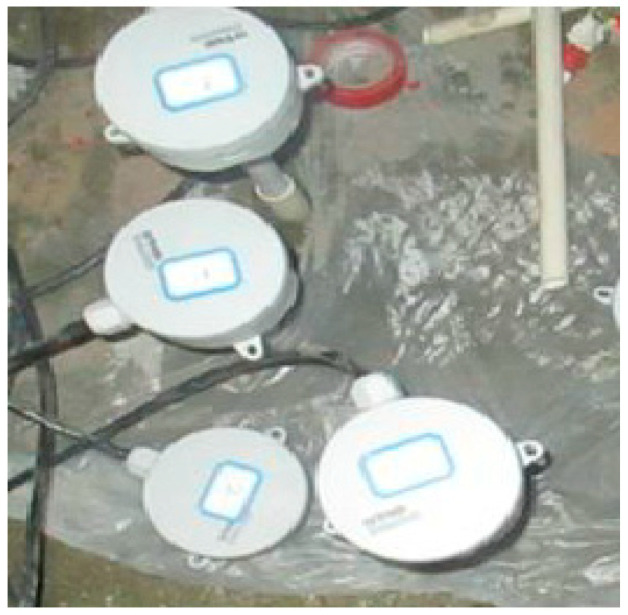
TCA-TIH resistive integrated temperature and humidity sensors.

**Figure 4 polymers-15-02207-f004:**
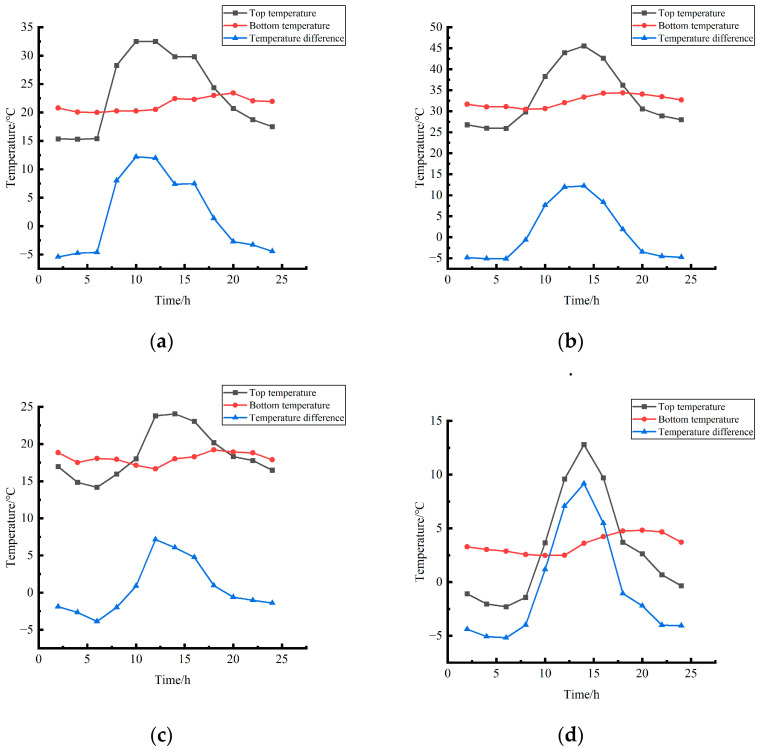
Average temperature field of pavement surface at different times in different seasons. (**a**) Spring. (**b**) Summer. (**c**) Autumn. (**d**) Winter.

**Figure 5 polymers-15-02207-f005:**
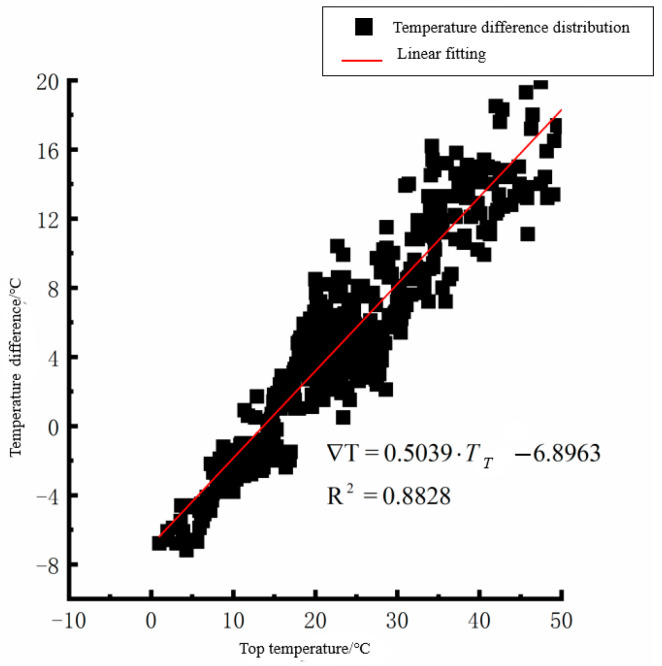
Top of plat temperature–temperature difference in spring.

**Figure 6 polymers-15-02207-f006:**
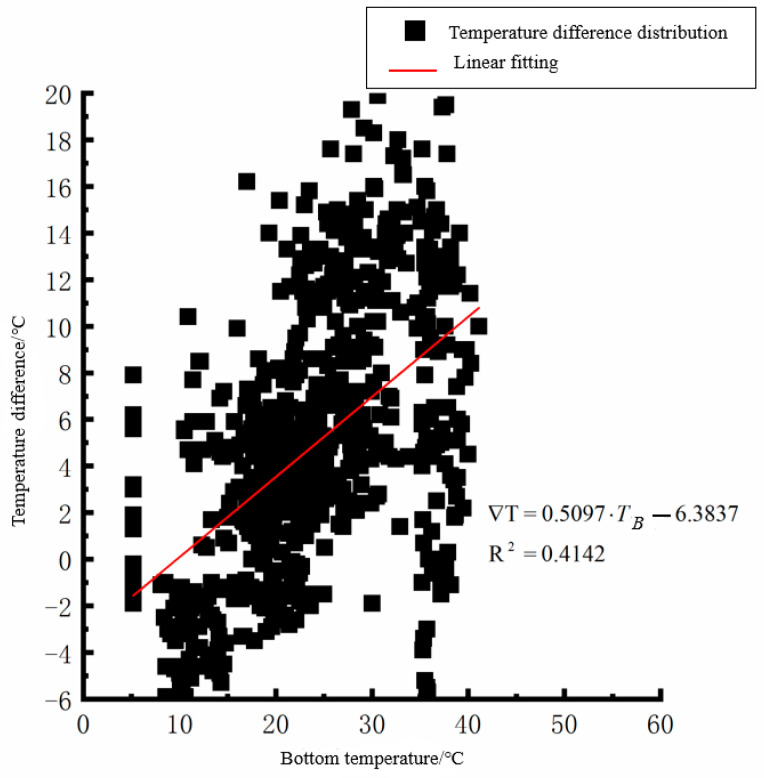
Bottom of plat temperature–temperature difference in spring.

**Figure 7 polymers-15-02207-f007:**
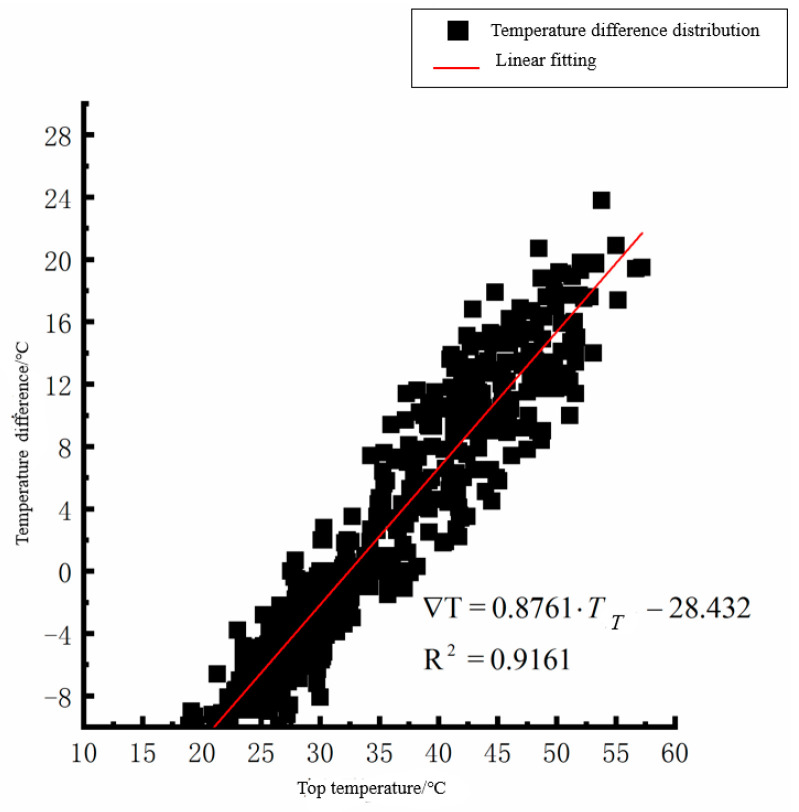
Top of plat temperature–temperature difference in summer.

**Figure 8 polymers-15-02207-f008:**
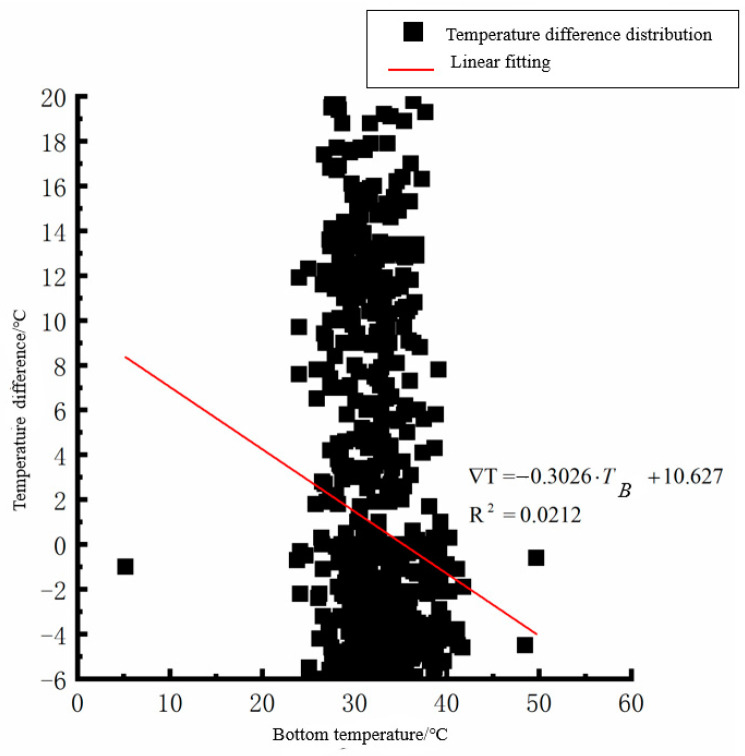
Bottom of plat temperature–temperature difference in summer.

**Figure 9 polymers-15-02207-f009:**
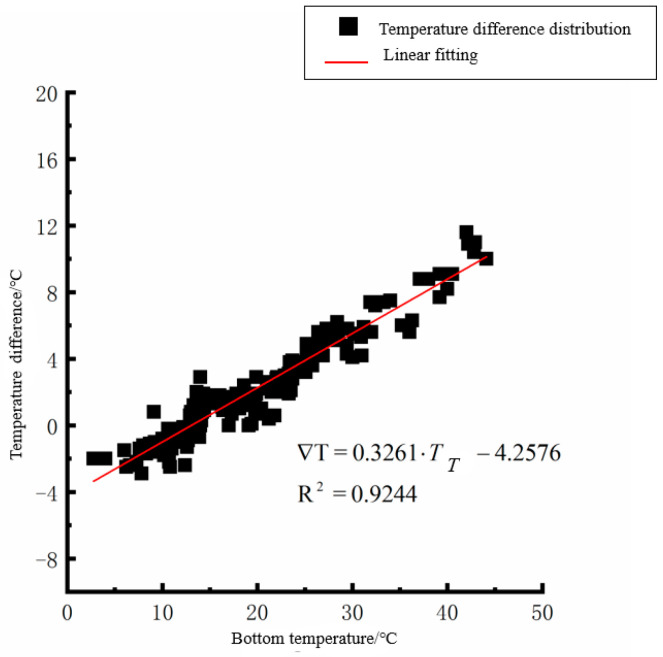
Top of plat temperature–temperature difference in autumn.

**Figure 10 polymers-15-02207-f010:**
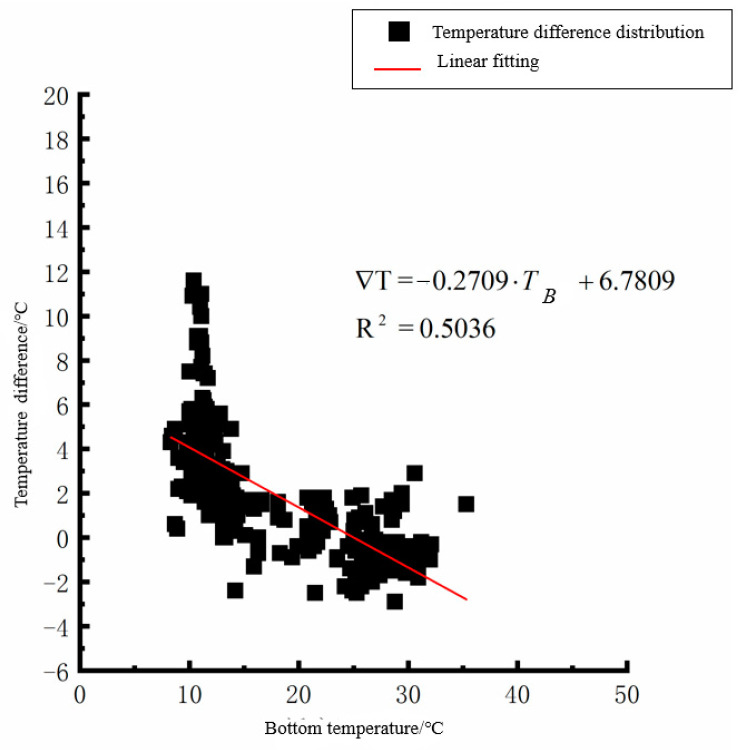
Bottom of plat temperature–temperature difference in autumn.

**Figure 11 polymers-15-02207-f011:**
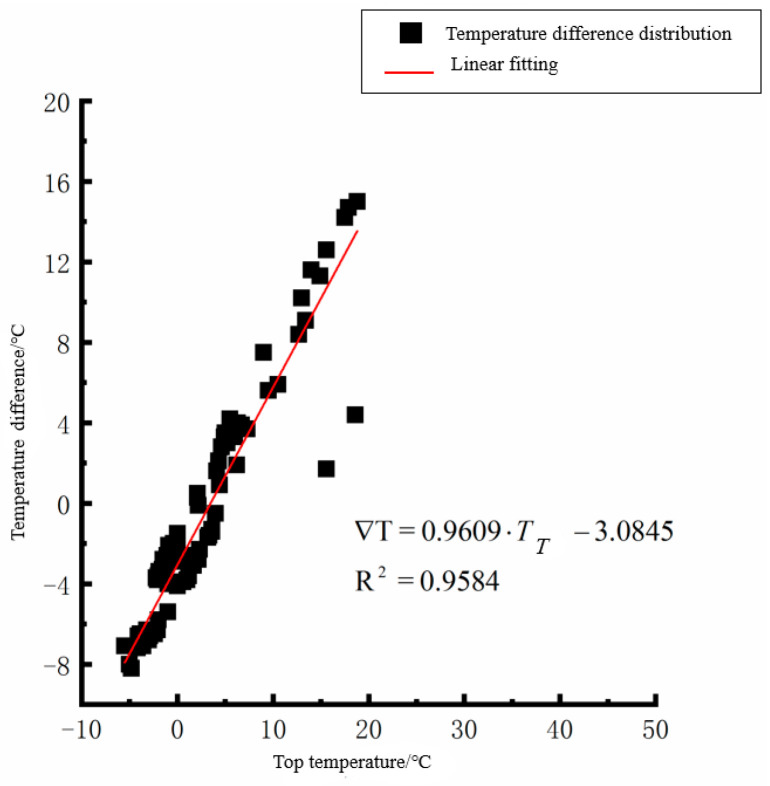
Top of plat temperature–temperature difference in winter.

**Figure 12 polymers-15-02207-f012:**
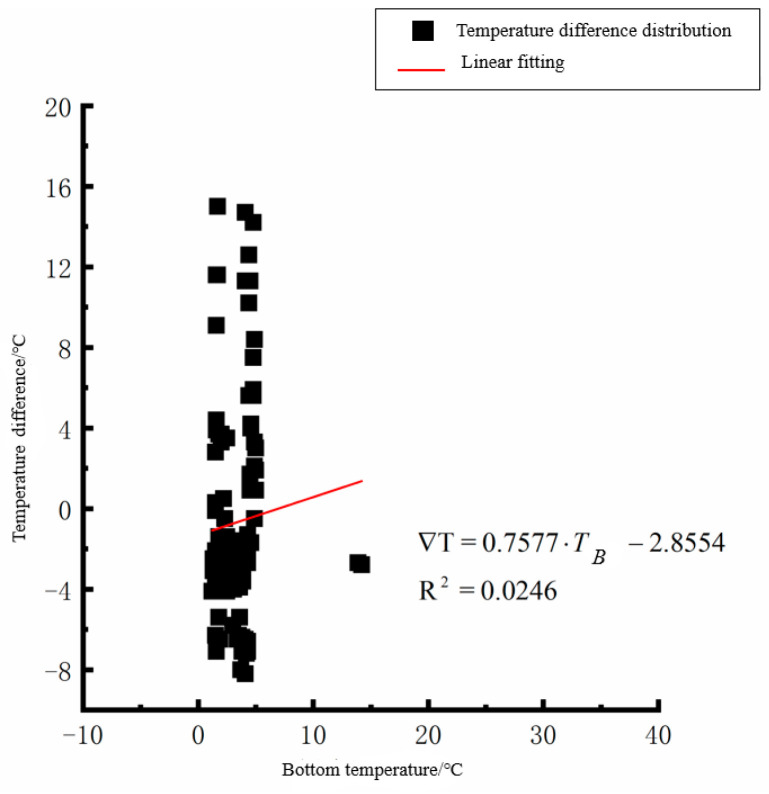
Bottom of plat temperature–temperature difference in winter.

**Figure 13 polymers-15-02207-f013:**
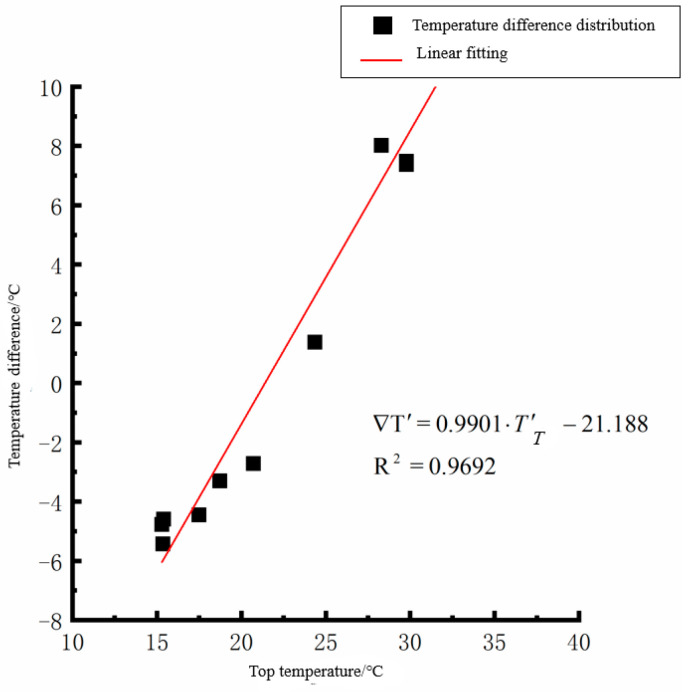
Average daily top temperature in spring—temperature difference.

**Figure 14 polymers-15-02207-f014:**
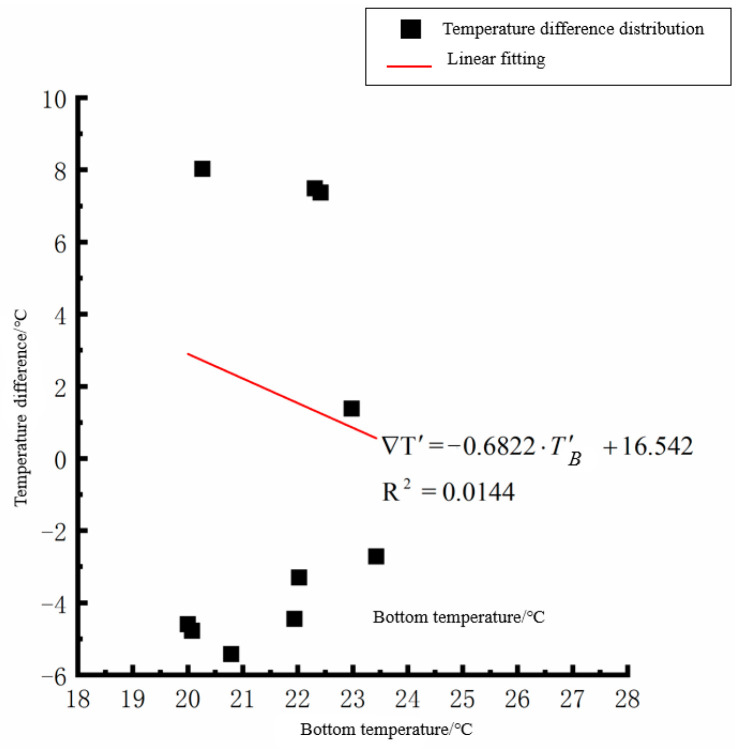
Average daily plate bottom temperature in spring—temperature difference.

**Figure 15 polymers-15-02207-f015:**
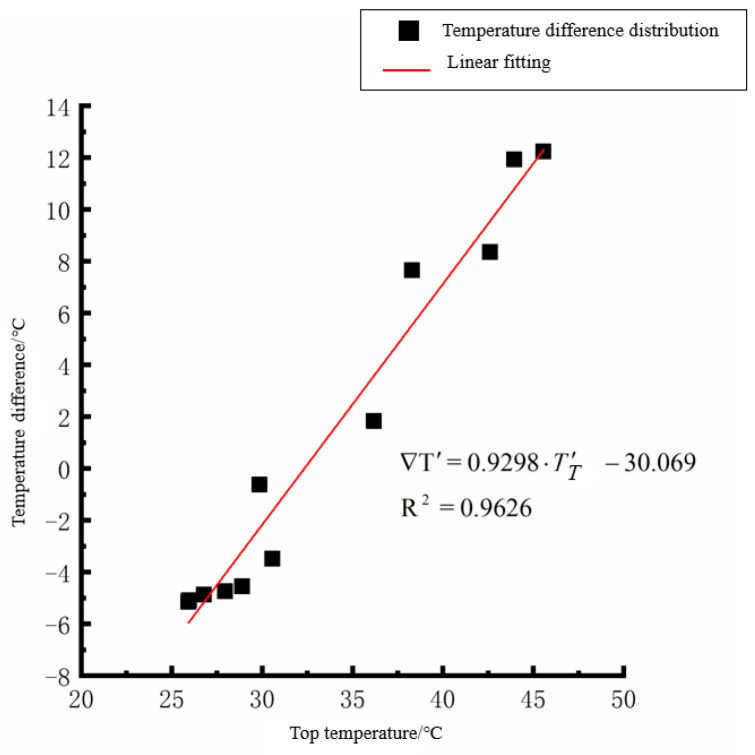
Average daily top temperature in summer—temperature difference.

**Figure 16 polymers-15-02207-f016:**
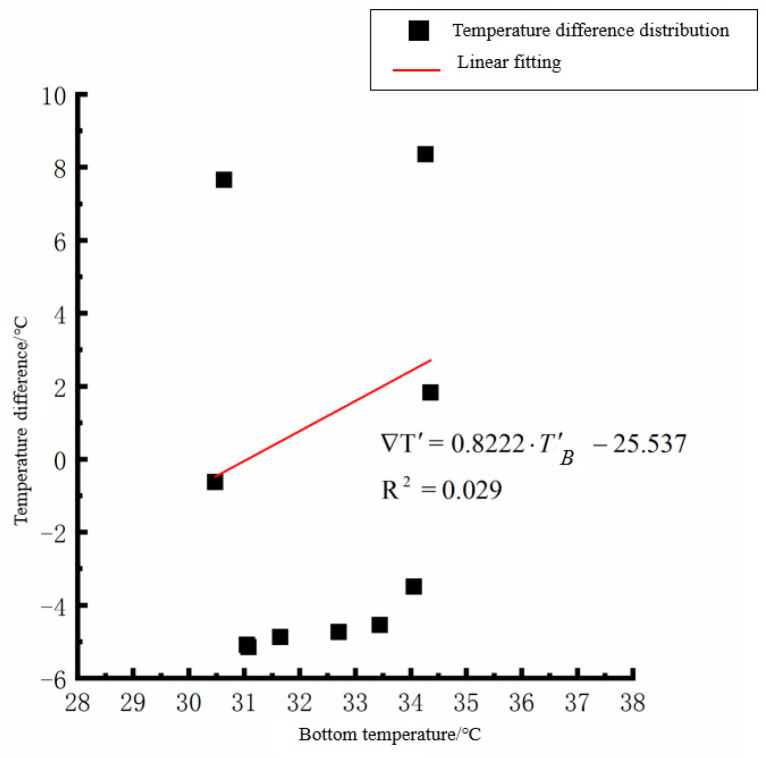
Average daily plate bottom temperature in summer—temperature difference.

**Figure 17 polymers-15-02207-f017:**
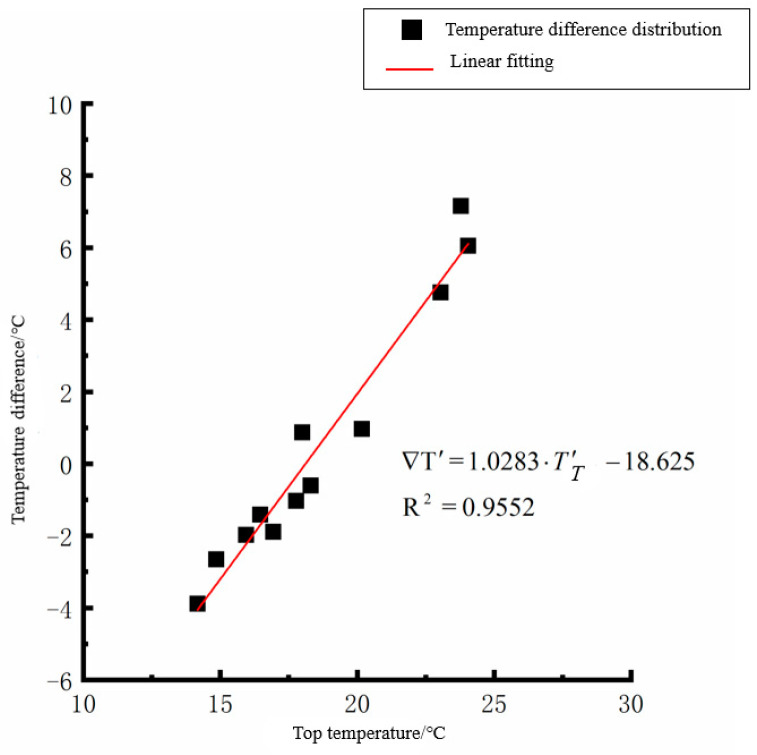
Average daily top temperature in autumn—temperature difference.

**Figure 18 polymers-15-02207-f018:**
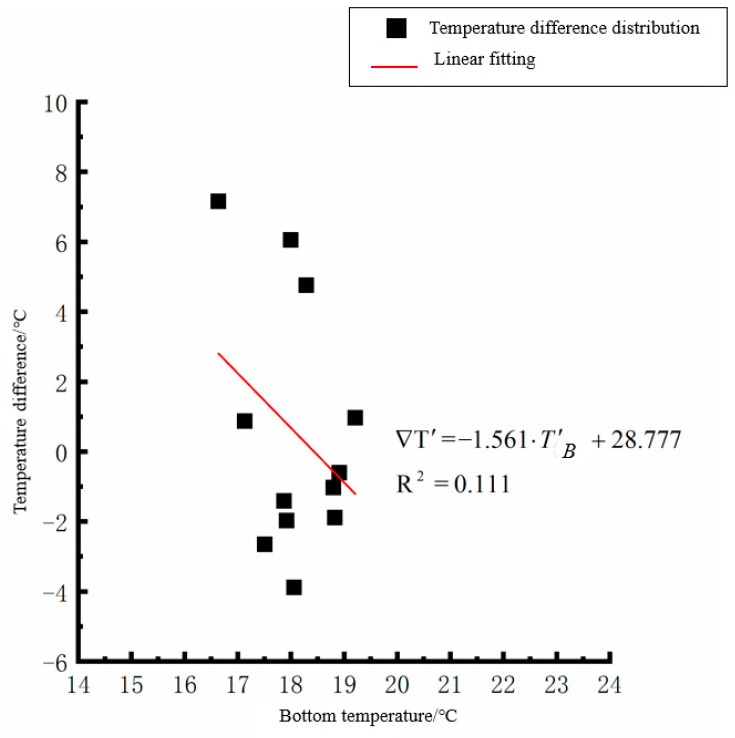
Average daily plate bottom temperature in autumn—temperature difference.

**Figure 19 polymers-15-02207-f019:**
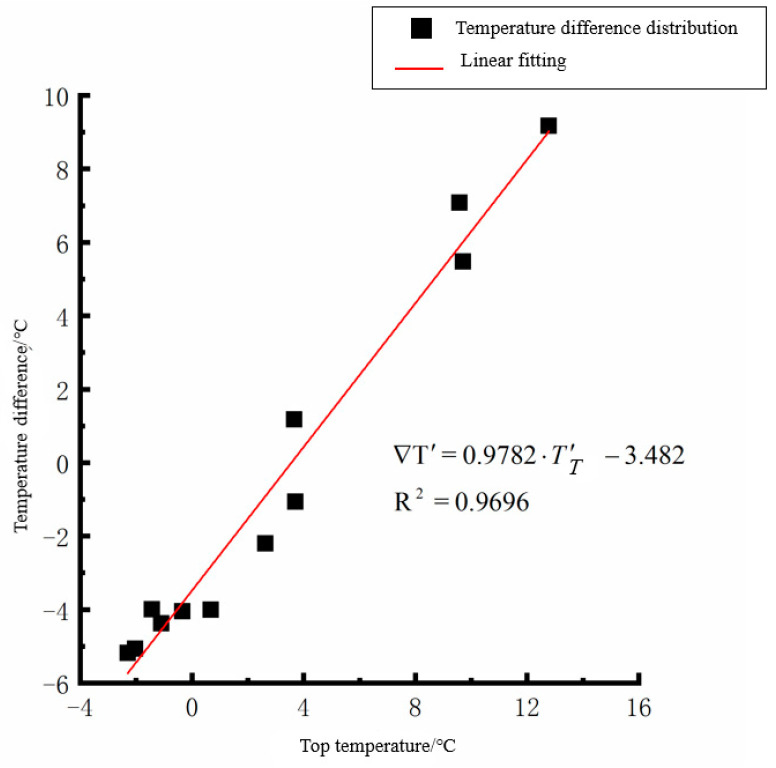
Average daily top temperature in winter—temperature difference.

**Figure 20 polymers-15-02207-f020:**
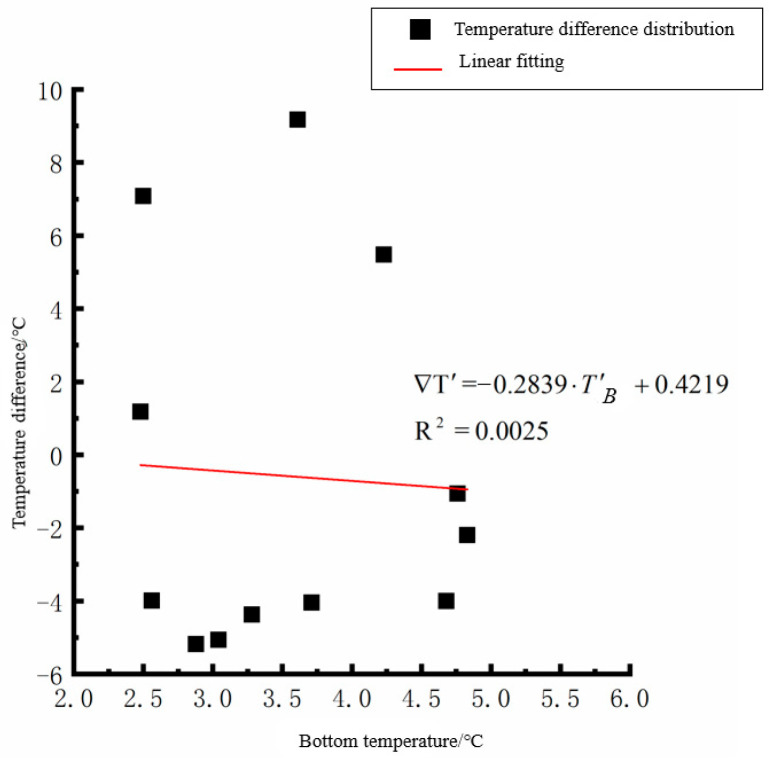
Average daily plate bottom temperature in winter—temperature difference.

**Figure 21 polymers-15-02207-f021:**
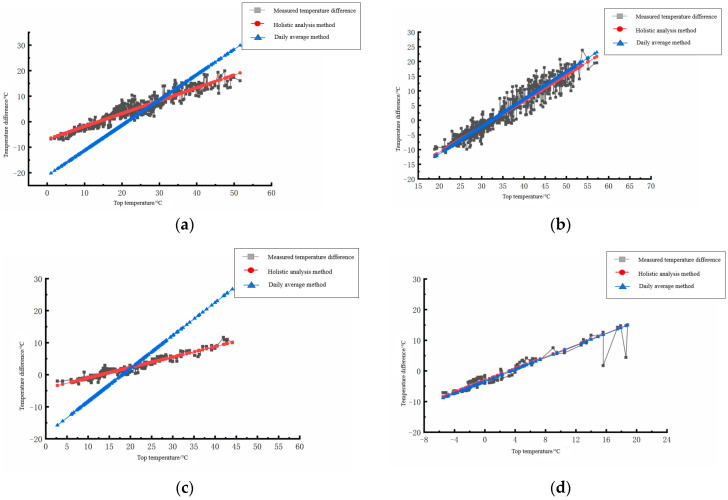
Comparison of the calculated and measured values of the two analysis methods in each season. (**a**) Comparison of correlation in spring. (**b**) Comparison of correlation in summer. (**c**) Comparison of correlation in autumn. (**d**) Comparison of correlation in winter.

**Figure 22 polymers-15-02207-f022:**
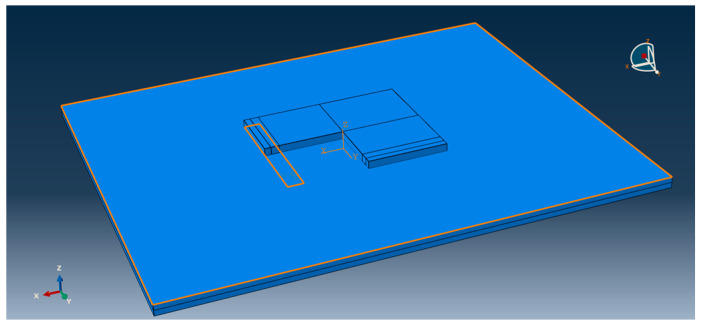
Cement pavement slab edge void (hidden 1/4 pavement slab).

**Figure 23 polymers-15-02207-f023:**
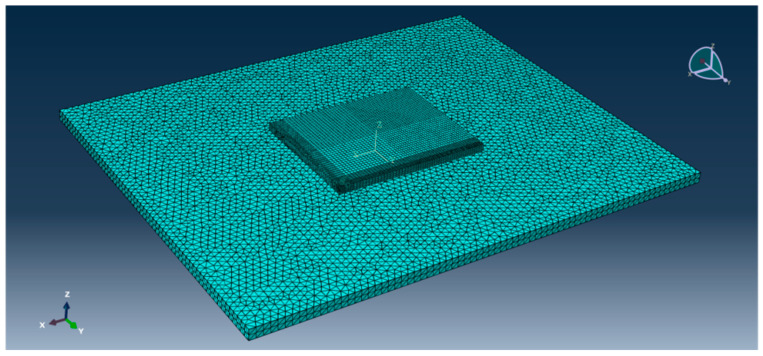
Overall mesh division effect of 3D finite element model of cement pavement.

**Figure 24 polymers-15-02207-f024:**
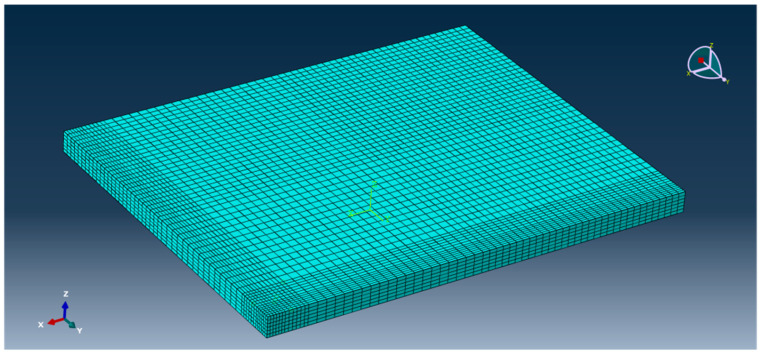
Mesh division effect of cement road panel model.

**Figure 25 polymers-15-02207-f025:**
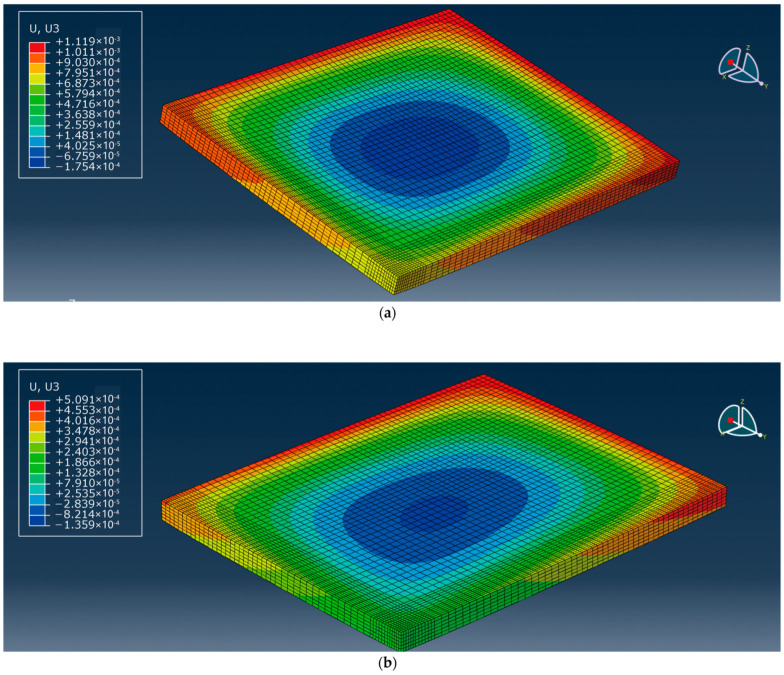
Deformation nephograph under different temperature stresses at the side of cement road panel. (**a**) Panel temperature difference −20 °C. (**b**) Panel temperature difference −10 °C. (**c**) Panel temperature difference 0 °C. (**d**) Panel temperature difference 10 °C. (**e**) Panel temperature difference 20 °C.

**Figure 26 polymers-15-02207-f026:**
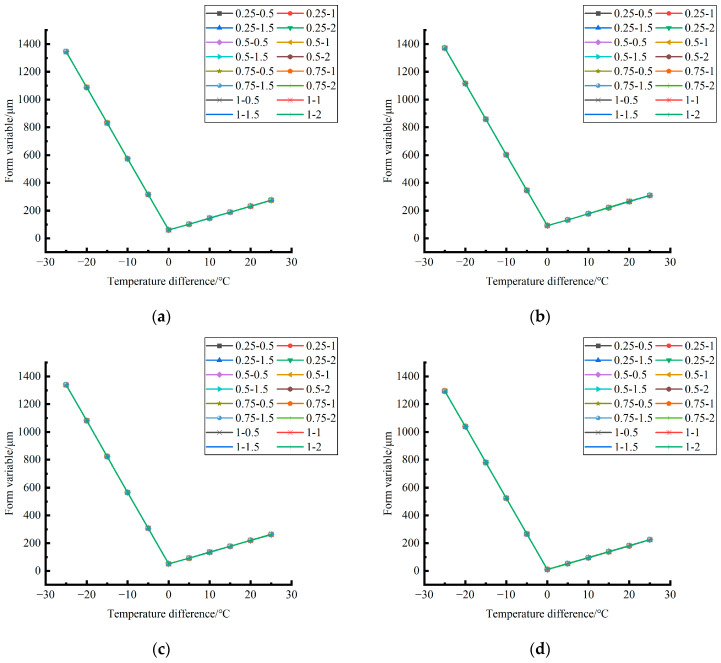
Deformation caused only by temperature stress in different seasons during plate edge hollowing. (**a**) Spring. (**b**) Summer. (**c**) Autumn. (**d**) Winter.

**Figure 27 polymers-15-02207-f027:**
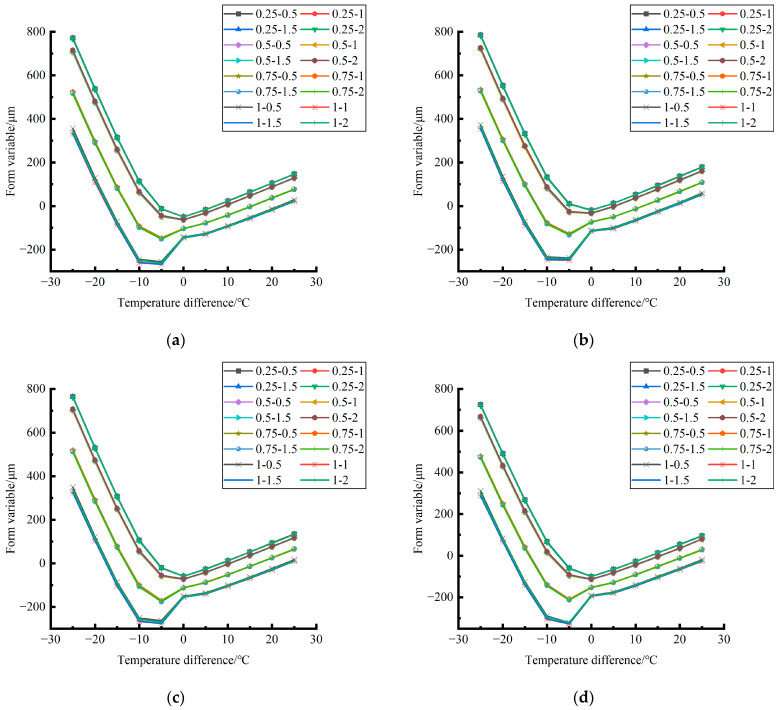
Deformation generated by coupling effect of temperature stress and load (50 KN) stress field in different seasons. (**a**) Spring. (**b**) Summer. (**c**) Autumn. (**d**) Winter.

**Figure 28 polymers-15-02207-f028:**
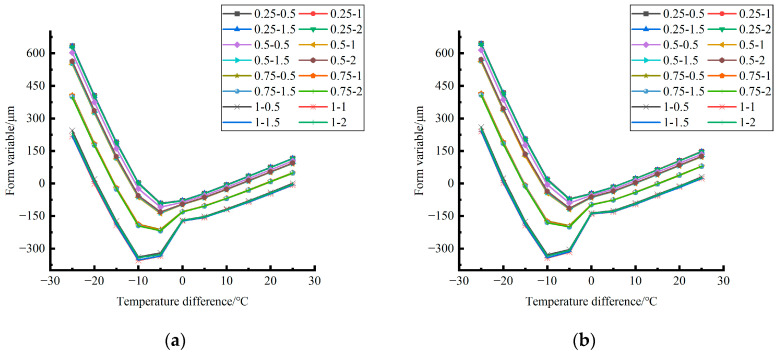
Deformation generated by coupling effect of temperature stress and load (70 KN) stress field in different seasons. (**a**) Spring. (**b**) Summer. (**c**) Autumn. (**d**) Winter.

**Figure 29 polymers-15-02207-f029:**
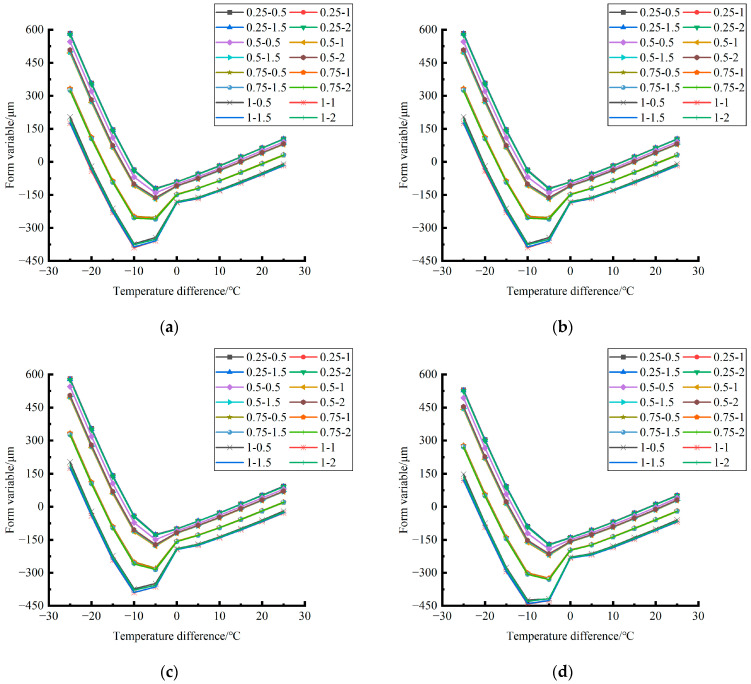
Deformation generated by coupling effect of temperature stress and load (90 KN) stress field in different seasons. (**a**) Spring. (**b**) Summer. (**c**) Autumn. (**d**) Winter.

**Figure 30 polymers-15-02207-f030:**
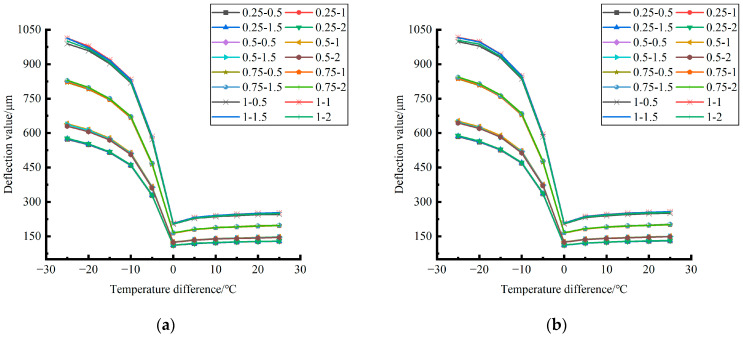
Bending considering the influence of temperature in different seasons (load 50 KN). (**a**) Spring. (**b**) Summer. (**c**) Autumn. (**d**) Winter.

**Figure 31 polymers-15-02207-f031:**
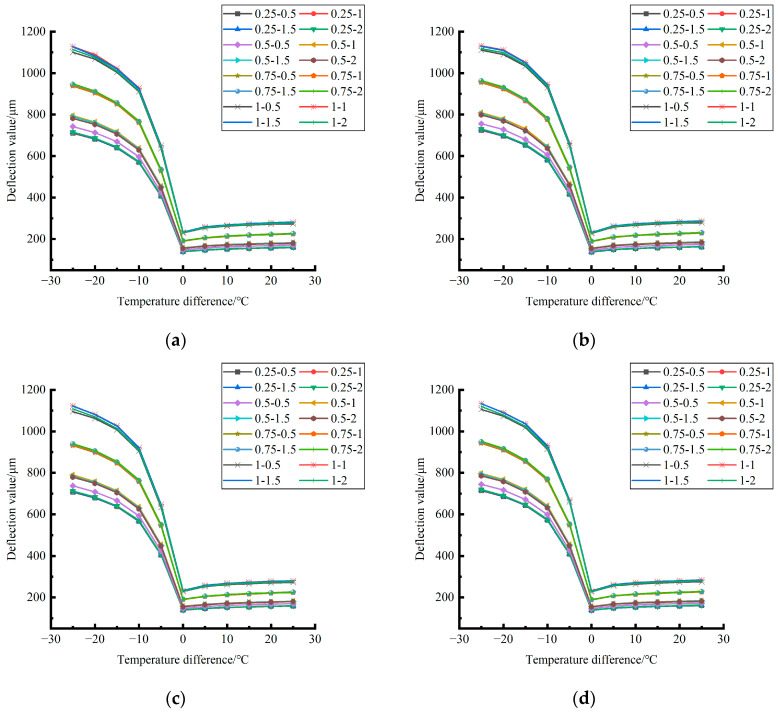
Bending considering the influence of temperature in different seasons (load 70 KN). (**a**) Spring. (**b**) Summer. (**c**) Autumn. (**d**) Winter.

**Figure 32 polymers-15-02207-f032:**
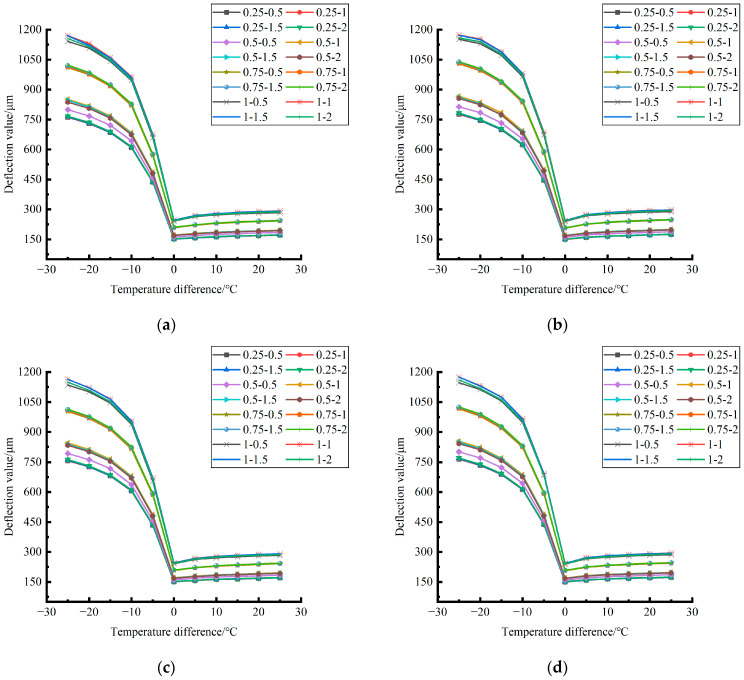
Bending under the influence of temperature in different seasons (load 90 KN). (**a**) Spring. (**b**) Summer. (**c**) Autumn. (**d**) Winter.

**Figure 33 polymers-15-02207-f033:**
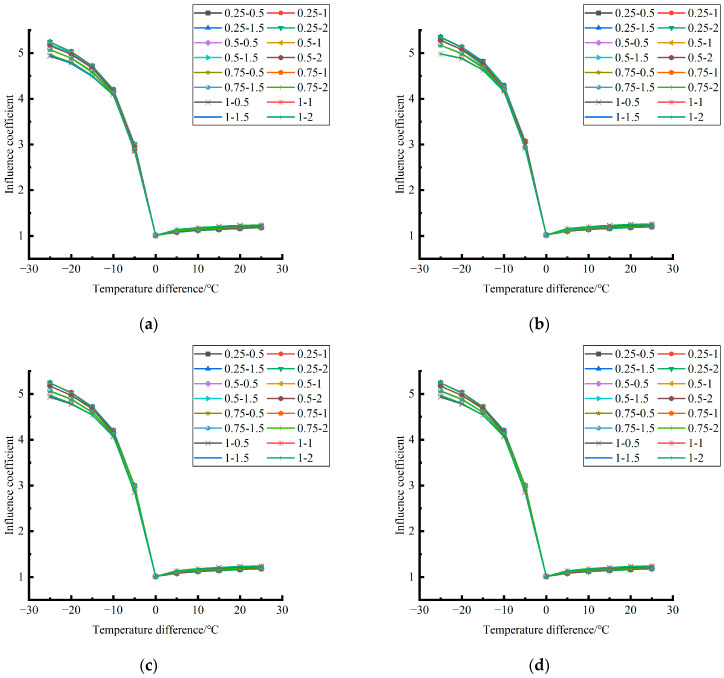
Influence coefficient of temperature in different seasons (load 50 KN). (**a**) Spring. (**b**) Summer. (**c**) Autumn. (**d**) Winter.

**Figure 34 polymers-15-02207-f034:**
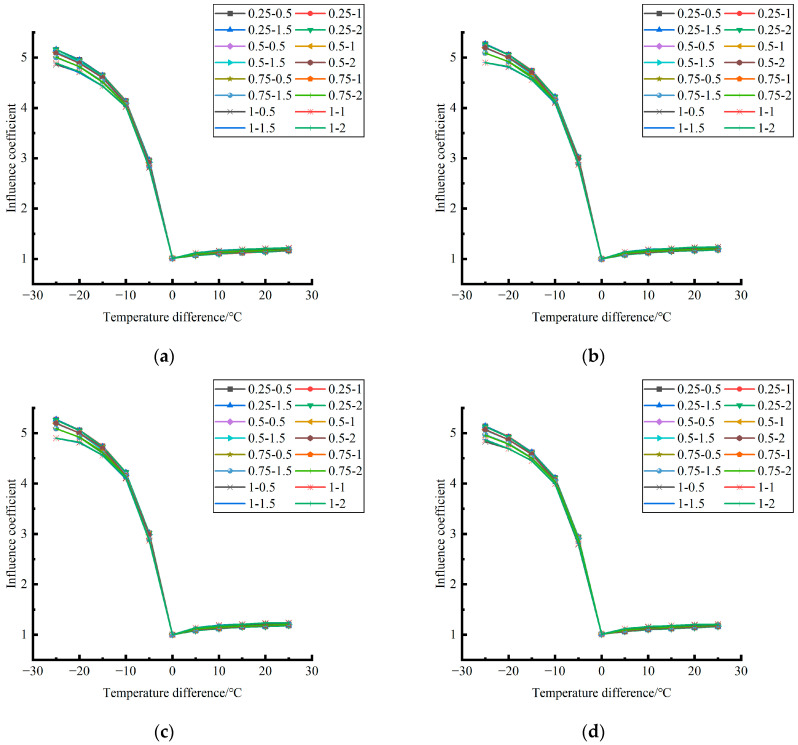
Influence coefficient of temperature in different seasons (load 70 KN). (**a**) Spring. (**b**) Summer. (**c**) Autumn. (**d**) Winter.

**Figure 35 polymers-15-02207-f035:**
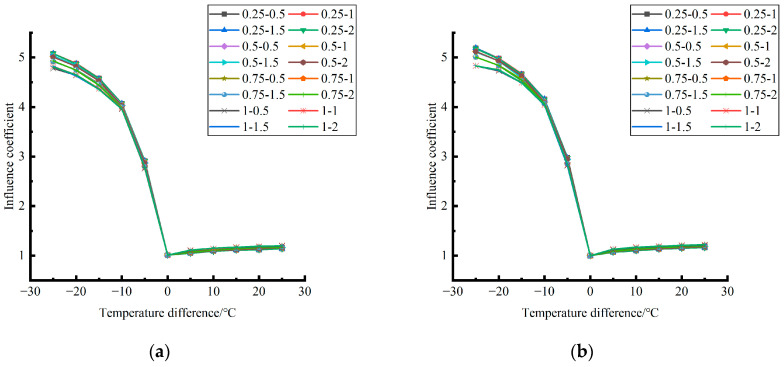
Influence coefficient of temperature in different seasons (load 90 KN). (**a**) Spring. (**b**) Summer. (**c**) Autumn. (**d**) Winter.

**Figure 36 polymers-15-02207-f036:**
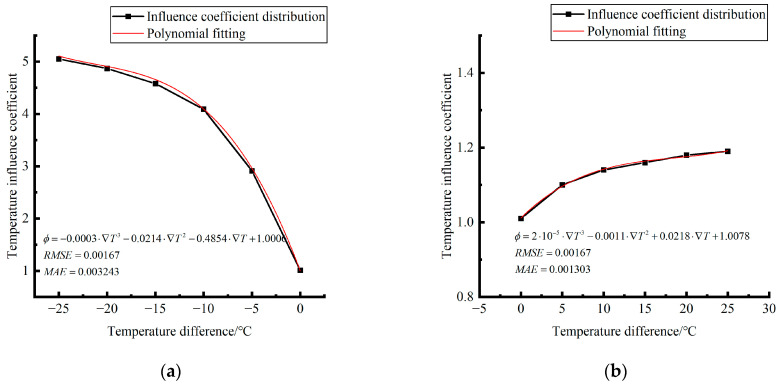
Correlation between positive and negative temperature differences and influence coefficient. (**a**) Range of negative temperature differences. (**b**) Range of positive temperature differences.

**Figure 37 polymers-15-02207-f037:**
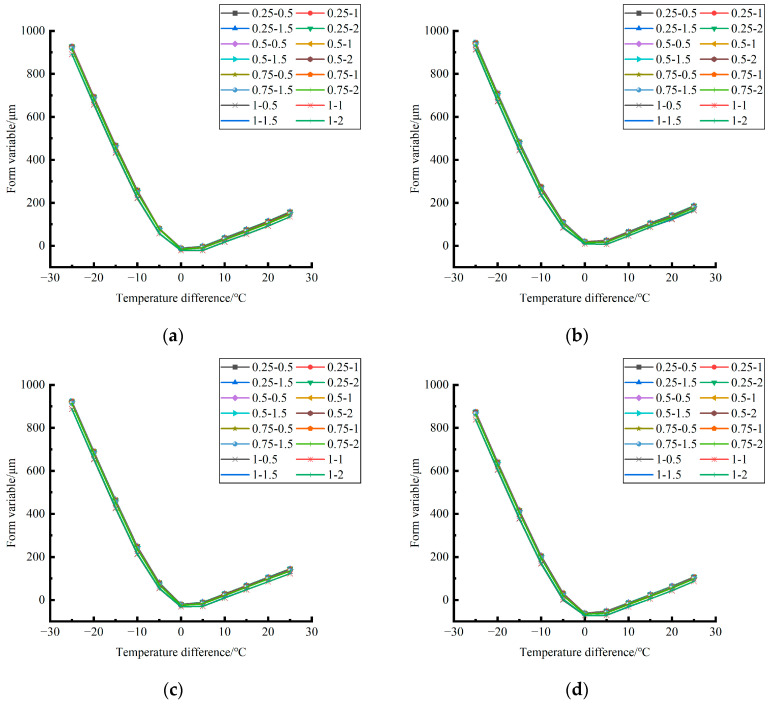
Deformation caused by the coupling of temperature stress and load (50 KN) stress field after plate edge cavity grouting in different seasons. (**a**) Spring. (**b**) Summer. (**c**) Autumn. (**d**) Winter.

**Figure 38 polymers-15-02207-f038:**
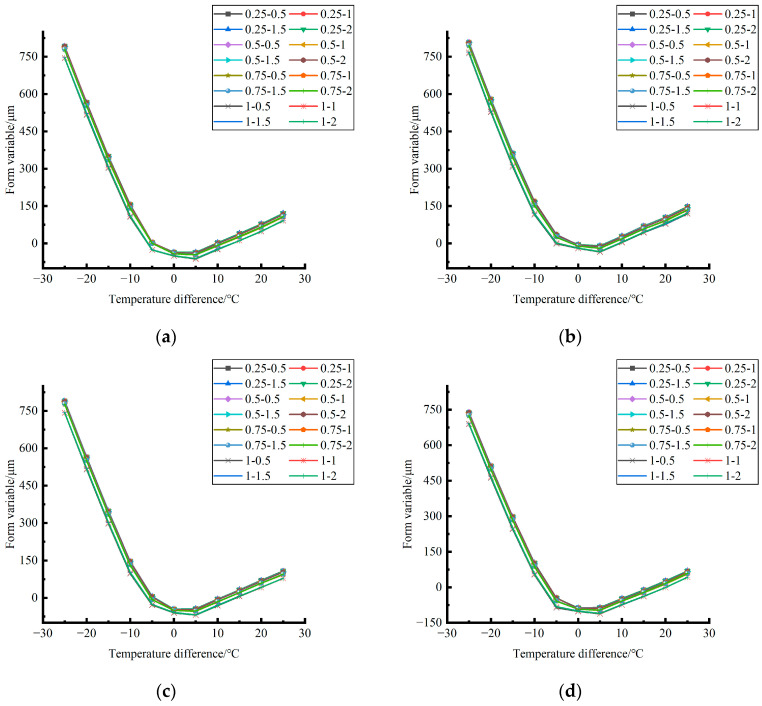
Deformation caused by the coupling of temperature stress and load (70 KN) stress field after plate edge cavity grouting in different seasons. (**a**) Spring. (**b**) Summer. (**c**) Autumn. (**d**) Winter.

**Figure 39 polymers-15-02207-f039:**
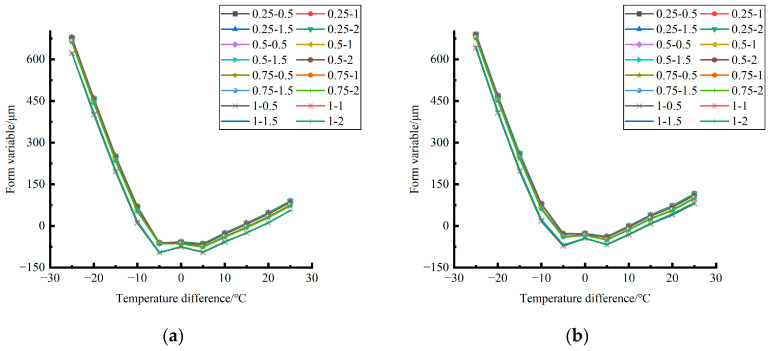
Deformation caused by the coupling of temperature stress and load (90 KN) stress field after angular cavity grouting in different seasons. (**a**) Spring. (**b**) Summer. (**c**) Autumn. (**d**) Winter.

**Figure 40 polymers-15-02207-f040:**
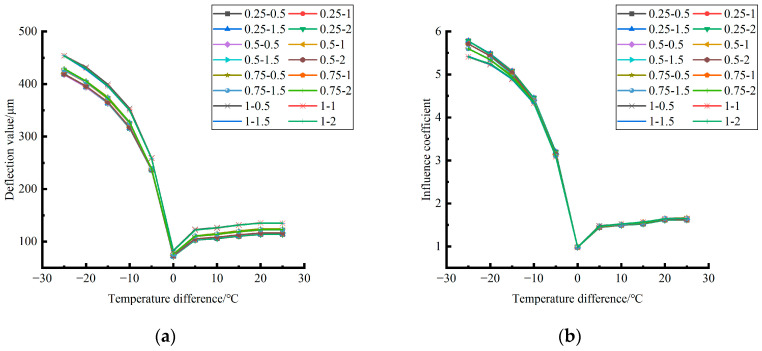
Bending considering the influence of temperature after plate angle unloading grouting in different seasons (load 50 KN). (**a**) Spring. (**b**) Summer. (**c**) Autumn. (**d**) Winter.

**Figure 41 polymers-15-02207-f041:**
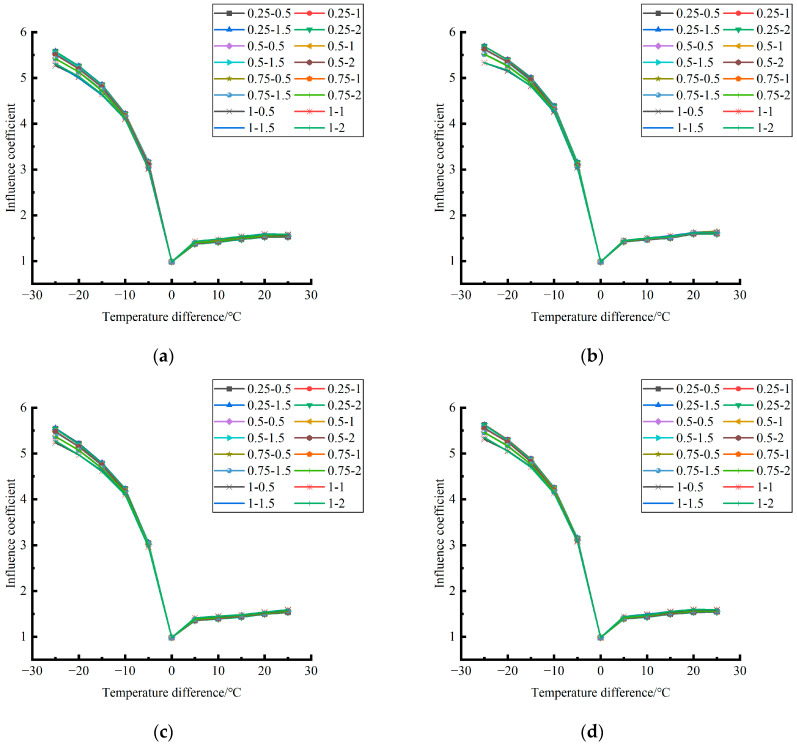
Bending of slab edge with temperature influence considered after hollow grouting in different seasons (load 70 KN). (**a**) Spring. (**b**) Summer. (**c**) Autumn. (**d**) Winter.

**Figure 42 polymers-15-02207-f042:**
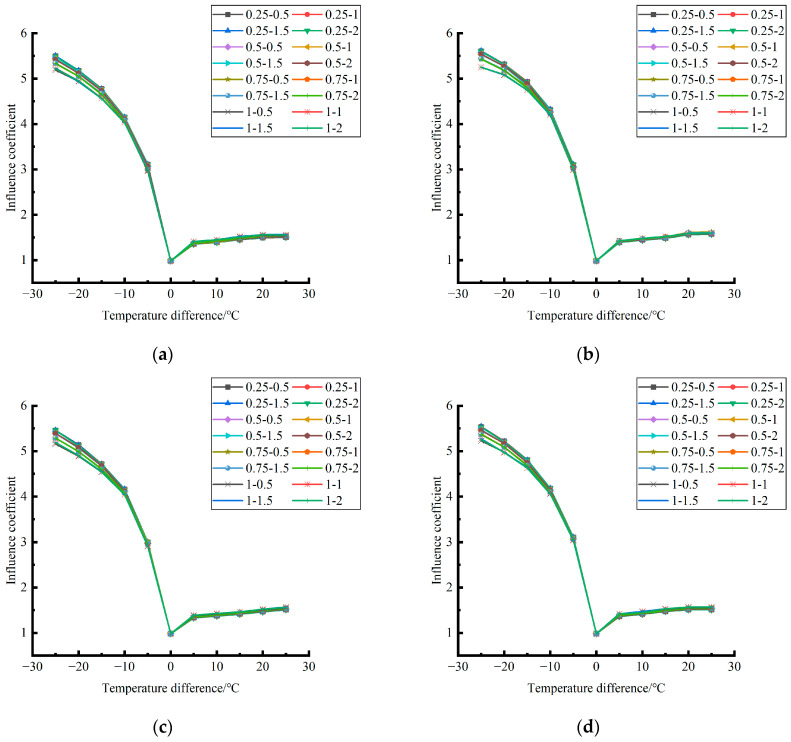
Deflection considering the influence of temperature after plate angle unloading grouting in different seasons (load 90 KN). (**a**) Spring. (**b**) Summer. (**c**) Autumn. (**d**) Winter.

**Figure 43 polymers-15-02207-f043:**
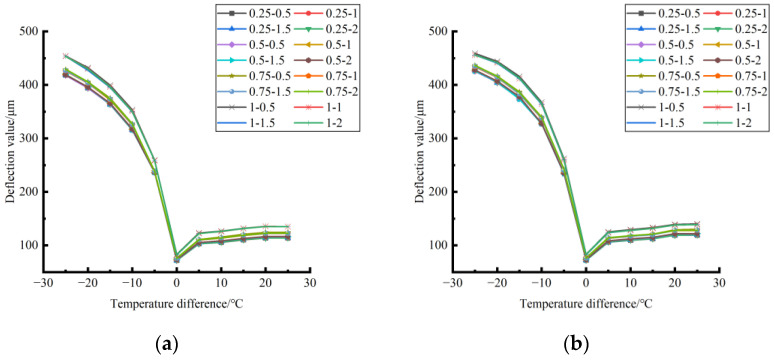
Influence coefficient of temperature after plate angle unloading grouting in different seasons (load 50 KN). (**a**) Spring. (**b**) Summer. (**c**) Autumn. (**d**) Winter.

**Figure 44 polymers-15-02207-f044:**
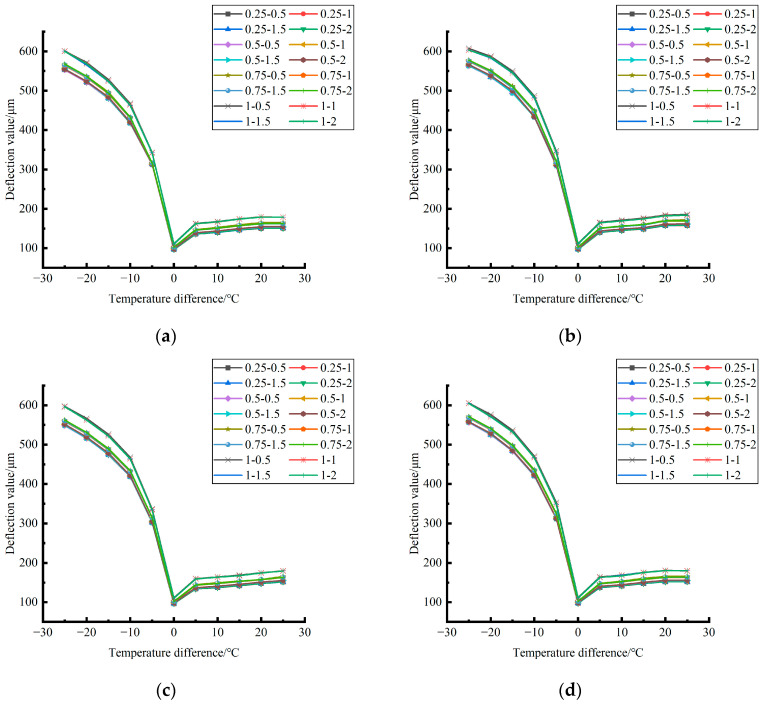
Influence coefficient of temperature after plate angle unloading grouting in different seasons (load 70 KN). (**a**) Spring. (**b**) Summer. (**c**) Autumn. (**d**) Winter.

**Figure 45 polymers-15-02207-f045:**
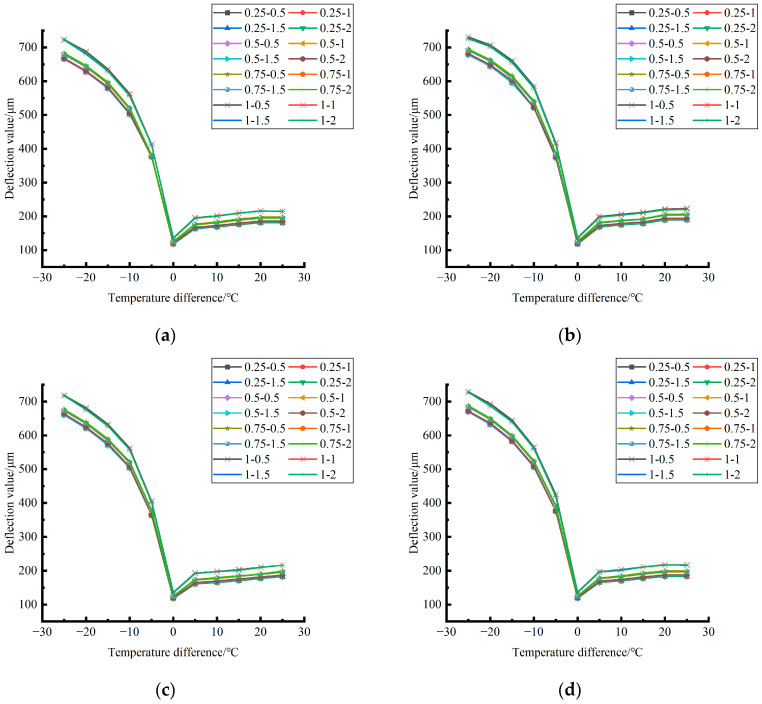
Influence coefficient of temperature after plate edge unloading grouting in different seasons (load 90 KN). (**a**) Spring. (**b**) Summer. (**c**) Autumn. (**d**) Winter.

**Figure 46 polymers-15-02207-f046:**
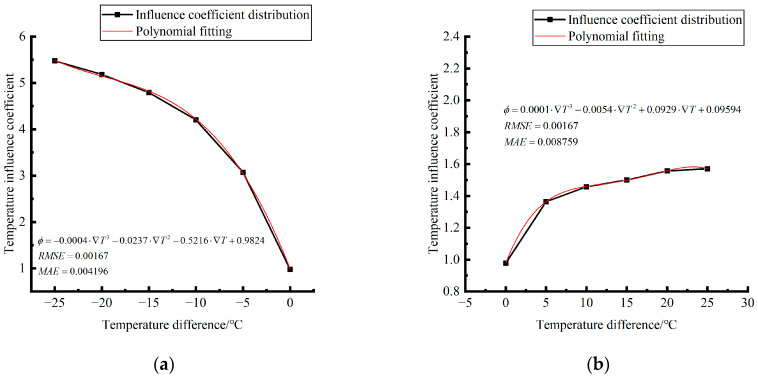
Correlation between positive and negative temperature differences and influence coefficient. (**a**) Range of negative temperature differences. (**b**) Range of positive temperature differences.

**Figure 47 polymers-15-02207-f047:**
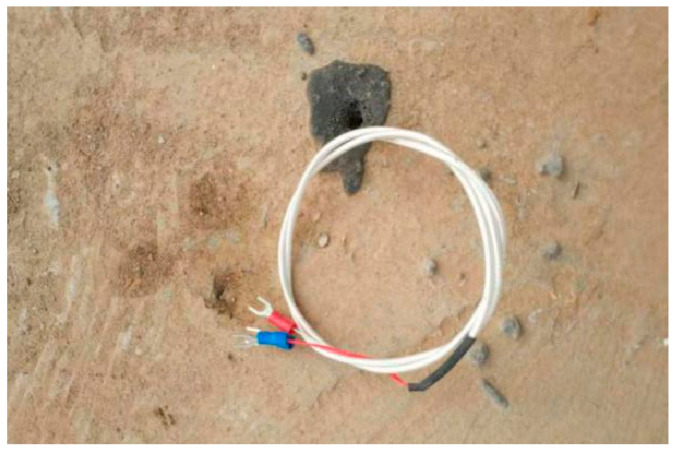
The temperature sensor was buried and blocked.

**Figure 48 polymers-15-02207-f048:**
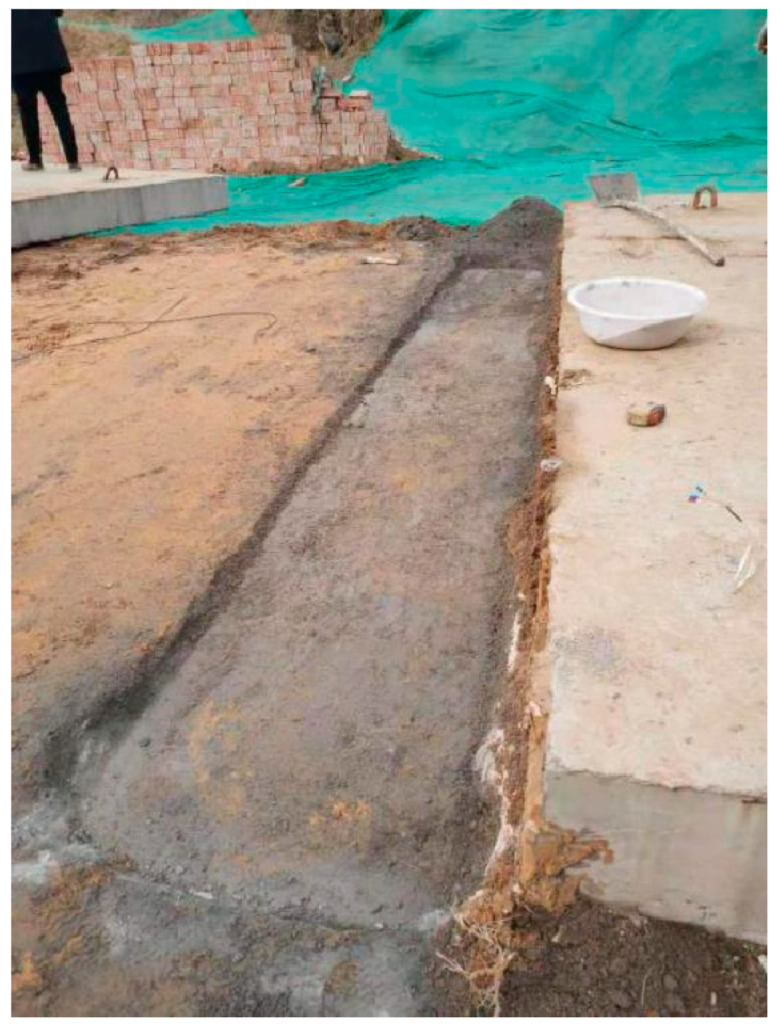
A void was arranged at the corresponding edge of the base plate.

**Figure 49 polymers-15-02207-f049:**
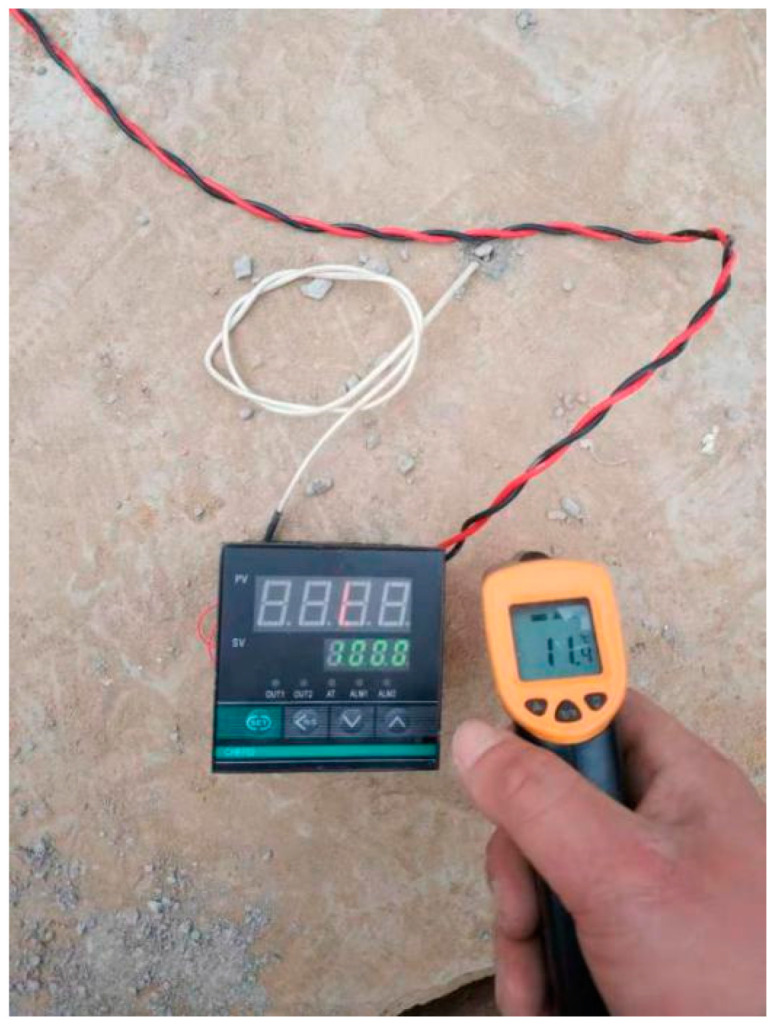
Measuring the pavement temperature.

**Figure 50 polymers-15-02207-f050:**
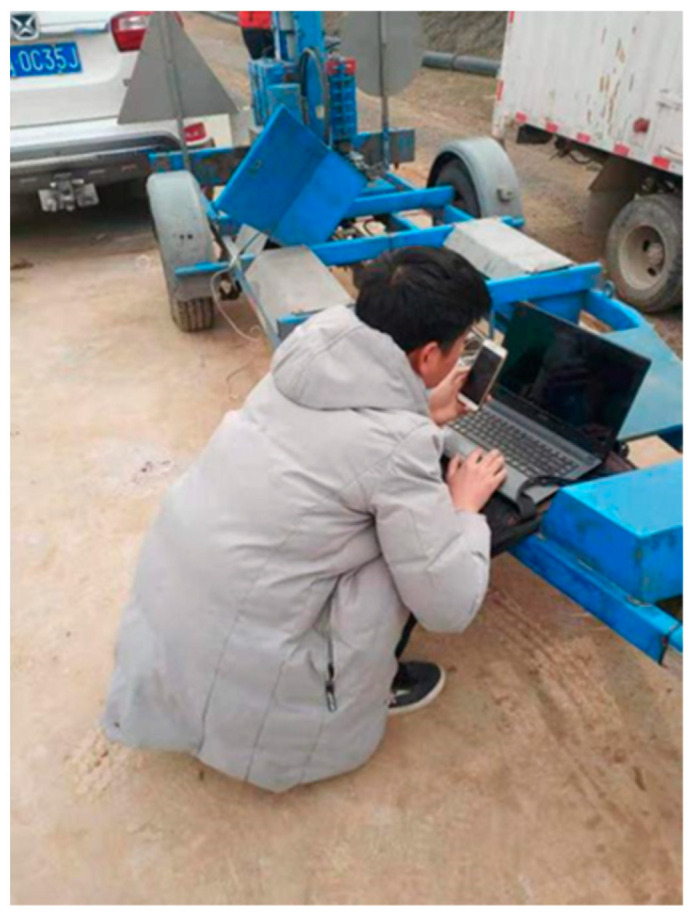
Measuring the bending data.

**Figure 51 polymers-15-02207-f051:**
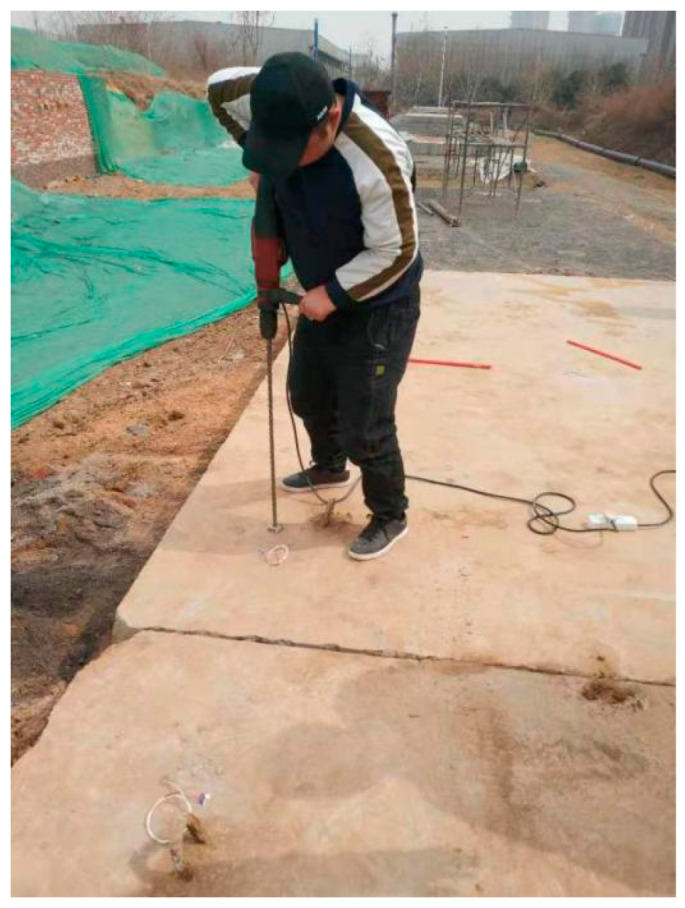
Drilling polymer grouting holes in the release area.

**Figure 52 polymers-15-02207-f052:**
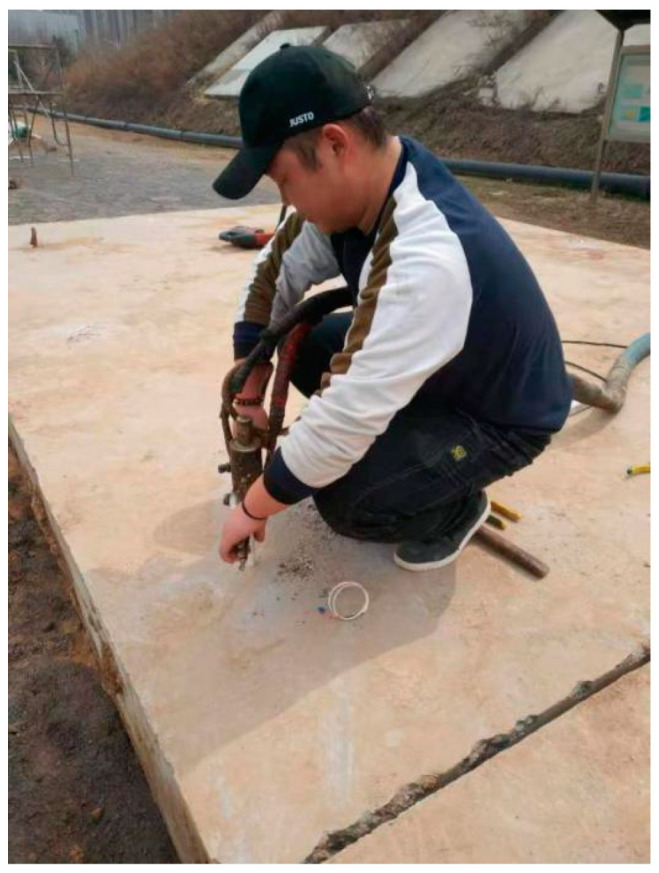
Polymer grouting in the void zone.

**Figure 53 polymers-15-02207-f053:**
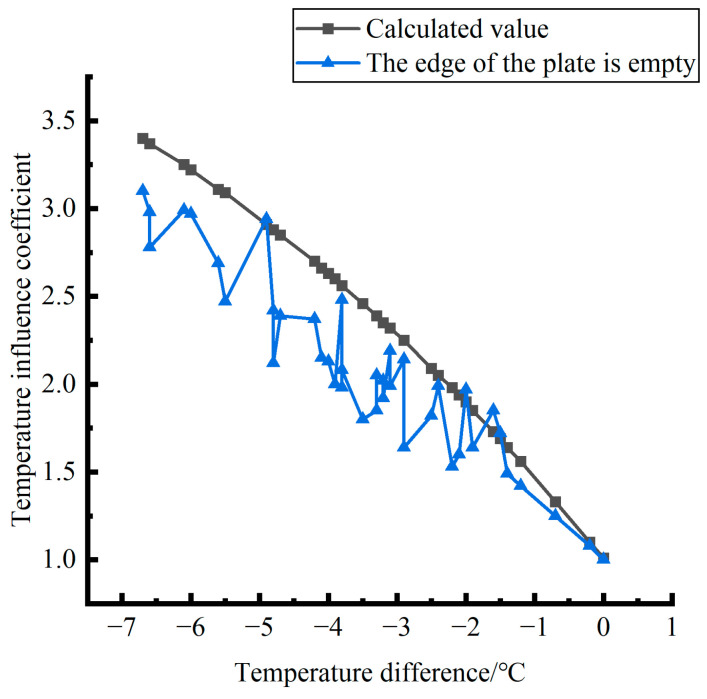
Range of negative temperature differences before grouting.

**Figure 54 polymers-15-02207-f054:**
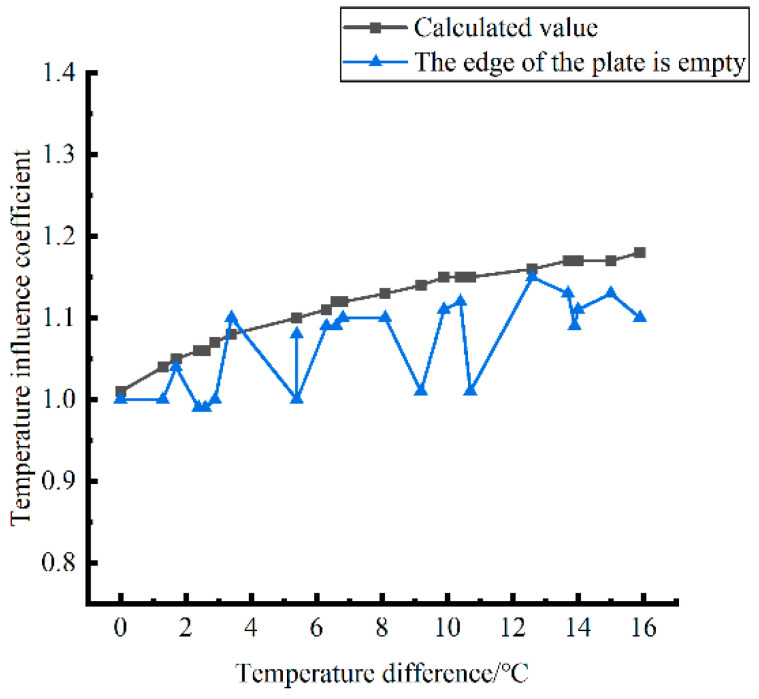
Range of positive temperature differences before grouting.

**Figure 55 polymers-15-02207-f055:**
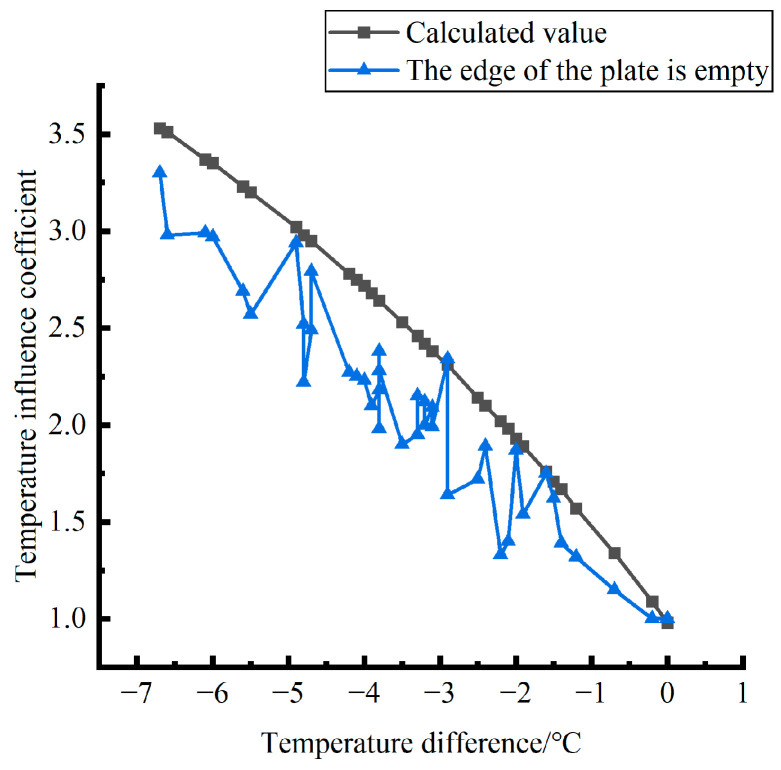
Range of negative temperature differences after grouting.

**Figure 56 polymers-15-02207-f056:**
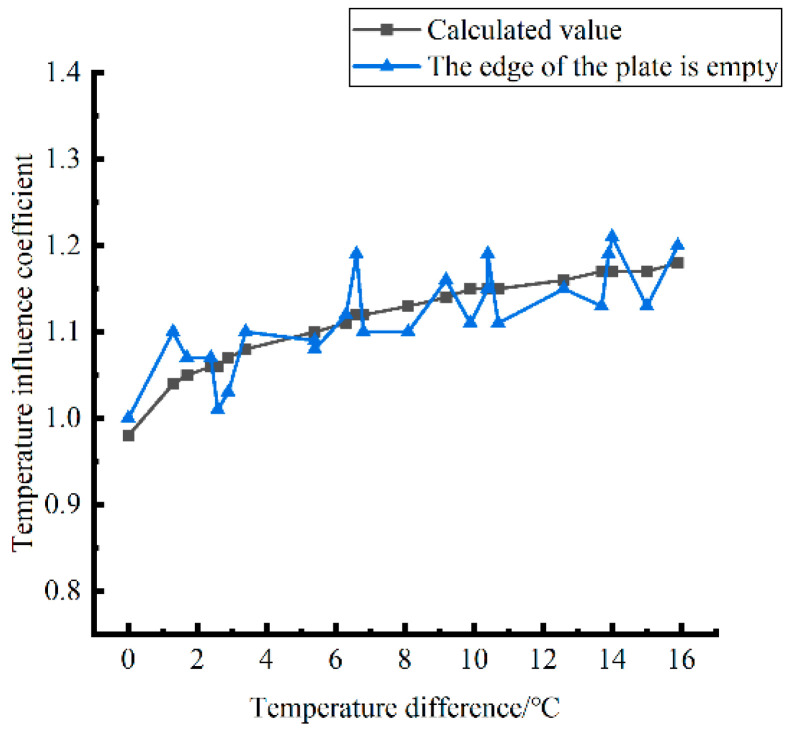
Range of positive temperature differences after grouting.

**Table 1 polymers-15-02207-t001:** Design parameters of each structural layer of cement concrete pavement.

Structural Layer	Surface Layer	Foundation Layer	Subbase Layer
Type of material	Cement Concrete	Cement-stabilized macadam base	Graded gravel
Thickness (dimension)	24 cm (4 m × 5 m)	18 cm	16 cm

**Table 2 polymers-15-02207-t002:** Material parameters of the cement concrete pavement lattice structure.

	Plane Size/m^2^	Thickness/cm	Modulus of Elasticity/MPa	Poisson Ratio	Thermal Conductivity/ *ω* m^−1^ °C^−1^	Coefficient of Expansion/°C^−1^
Surface layer	5 × 4	24	30,000	0.15	1.34	1.00 × 10^−5^
Foundation layer	15 × 12	16	500	0.3	-	-
Subbase layer	15 × 12	18	400	0.3	-	-

## Data Availability

All data, models, and code generated or used during the study appear in the published article.

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
