# Peer review of "Identification of Cement Pavement with Temperature Effect and Evaluation of Polymer Grouting Effect"

_polymers, 2023, doi:10.3390/polym15092207_

Round 1
Reviewer 1 Report
1. Please highlight the creative work in abstract and introduction. The logic in the two parts should be carefully considered.
2. Please check the tense in abstract. Both past and present tenses have been used to declare the similar content. Please make them in accordance.
3. Please check the stucture of the major content.
4. The quality of all the pictures in the whole manuscript should be improved.
5. Please declare the creative work in the manuscript. Please highlight the contribution of the study.
6. The English should be checked and improved. Please check the whole manuscript.
7. The conclusion part should be refined. The major findings should be expressed.
Author Response
Journal: Polymers
Manuscript ID: polymers-2311460
Manuscript title: Identification of cement pavement with temperature effect and evaluation of polymer grouting effect
Author: Xifeng Du, Haoyuan Cheng, Shengjie Xu and Wenjun Pei
Corresponding author: Xifeng Du, 202012222014206@gs.zzu.edu.cn
Dear Reviewer,
Thanks for your comments on our article " Identification of cement pavement with temperature effect and evaluation of polymer grouting effect " (ID: polymers-2311460). These opinions greatly value our paper's revision and improvement and have important guiding significance for our research. We have carefully studied these comments and tried to revise and improve the manuscript. We would like your approval.
We numbered the reviewer's suggestions and comments in italics for ease of differentiation. The primary corrections and responses to the reviewer's comments are listed below. All the changes are included in the new manuscript. To compare traces of change, please upload a copy of the manuscript for reference.
Thank you again for your time and consideration. Looking forward to hearing from you.
Sincerely yours,
Xifeng Du
Corresponding author
Responds to the reviewer’s comments:
- Response to comment: Please highlight the creative work in abstract and introduction. The logic in the two parts should be carefully considered.
Response: Thank you very much indeed for your comments. We have rewritten and adjusted the abstract and introduction logically better to explain our research's key content and results.
- Response to comment: Please check the tense in abstract. Both past and present tenses have been used to declare the similar content. Please make them in accordance.
Response: Thank you very much indeed for your comments. We have modified the tenses in the abstract better to state our research's actual contents and conclusions.
- Response to comment: Please check the structure of the major content.
Response: Thank you very much indeed for your comments. We have adjusted some parts of the paper, and the overall content structure is as follows :1. The change law of the temperature field is studied; 2. We carried out the cavity simulation under various working conditions according to the variation law of the obtained temperature field and the identification method of cement pavement. 3. Carried out the evaluation of the polymer grouting effect considering the temperature effect put forward and carried out the test.
- Response to comment: The quality of all the pictures in the whole manuscript should be improved.
Response: Thank you very much indeed for your comments. We have improved the quality of the pictures in the article by using higher pixel pictures to ensure clearer pictures and higher printing quality.
- Response to comment: Please declare the creative work in the manuscript. Please highlight the contribution of the study.
Response: Thank you very much indeed for your comments. We rewrote the paper's abstract, introduction, and conclusion to present our research results and contributions better.
- Response to comment: The English should be checked and improved. Please check the whole manuscript.
Response: Thank you very much indeed for your comments. We have asked Dr. Bei Zhang, a well-established expert, to polish our paper. Please see if the revised version met the English presentation standard.
- Response to comment: The conclusion part should be refined. The major findings should be expressed.
Response: Thank you very much indeed for your comments. We summarize the paper's conclusion and further discuss the application of the research content to literature and engineering. Your suggestions significantly improve this article.

Reviewer 2 Report
* Figures should be more legible and print quality should be high.
* Comment on the reasons for the correlation difference between equations 1-4
* In the conclusion, highlight the contributions of the study to the literature. It is not enough to interpret the results obtained.
Author Response
Journal: Polymers
Manuscript ID: polymers-2311460
Manuscript title: Identification of cement pavement with temperature effect and evaluation of polymer grouting effect
Author: Xifeng Du, Haoyuan Cheng, Shengjie Xu and Wenjun Pei
Corresponding author: Xifeng Du, 202012222014206@gs.zzu.edu.cn
Dear Reviewer,
Thanks for your comments on our article " Identification of cement pavement with temperature effect and evaluation of polymer grouting effect " (ID: polymers-2311460). These opinions greatly value our paper's revision and improvement and have important guiding significance for our research. We have carefully studied these comments and tried to revise and improve the manuscript. We would like your approval.
We numbered the reviewer's suggestions and comments in italics for ease of differentiation. The primary corrections and responses to the reviewer's comments are listed below. All the changes are included in the new manuscript. To compare traces of change, please upload a copy of the manuscript for reference.
Thank you again for your time and consideration. Looking forward to hearing from you.
Sincerely yours,
Xifeng Du
Corresponding author
Responds to the reviewer’s comments:
- Response to comment: Figures should be more legible and print quality should be high.
Response: Thank you very much indeed for your comments. We have improved the quality of the figures in the article by using higher pixel figures to ensure clearer figures and higher printing quality.
- Response to comment: Comment on the reasons for the correlation difference between equations 1-4.
Response: Thank you very much indeed for your comments. We add the notes in Equations 1-4 and the reasons for the correlation difference, ’ where: ∇T is the temperature difference at the bottom of the pavement slab roof, ℃; is pavement roof temperature, ℃. As seen from Figure 5-12, the TT coefficients in Equations (1)-(4) differ due to the different average temperatures of the plate roof in each season.’ (Lines 148 to 151)
- Response to comment: In the conclusion, highlight the contributions of the study to the literature. It is not enough to interpret the results obtained.
Response: Thank you very much indeed for your comments. We summarize the paper's conclusion and further discuss the application of the research content to literature and engineering. Your suggestions significantly improve this article.

Reviewer 3 Report
This work illustrates the correlation between temperature and pavement deformation. There are many simulation data displayed while limited experimental data is displayed except for the temperature.
General comment: authors should consider including the experimental deformation data to demonstrate the feasibility of the model. Without the experimental data, any conclusion drawn from this study is not justified. In addition, there are many grammar mistakes and authors should consider seeking for professional editing help.
Specific questions:
1. It appears that there is no correlation between the bottom temperature and temperature difference across all seasons and these data were not used in any further analysis. What is the purpose of analyzing it?
2. Figs 5, 7, 9, 11 plotted the temperature difference versus top temperature while description in line 135, 137, 139, 141 has no reference object. For example, we usually say ‘correlation between A and B is demonstrated…’ but description in line 135, 137, 139, 141 has the format as ‘correlation between A is…’. It appears as a grammar mistake but if I understand incorrectly, please correct me.
3. Line 198-209 listed 3 assumptions for developing multi-layer model. The assumption # 3 indicates that there is perfect contact between two layers, while in reality, defects in contact exist universally. What would be the ultimate effect if we have imperfect contact? My understanding is that stress concentration will occur, and the model author developed will not be valid anymore. Can authors comment on that?
4. The mesh density that authors used was 0.2. Why do authors choose this value? What will happen if we choose mesh density that is higher or lower than 0.2?
5. Legends from Fig25 to Fig45 are not well defined. For example, Figure 26 only shows data in one color but the legends indicated there are 16 colored data. Same thing happens to all rest of the figures. Authors should fix this.
6. Illustrations of Equation 11 from line 316 to line 320 indicates that Ltemperature stress is the combination deformation while Ltemperature stress and load is the deformation under temperature stress only. The subscript does not align with the description. Authors should double check.
7. Equations 13, 14,15,16 shows the correlation between influence coefficient and temperature difference. It very strange that authors displace R2 value for these equations since the equations are clearly not demonstrating linear correlation. It is the basic statistical analysis knowledge that R2 can only be used in linear correlation and the display of R2 here is scientifically wrong. Authors can do chi squared test or other statistical analysis instead.
8. What is DelT0 and DelT in Equations 13, 14,15,16?
Author Response
Journal: Polymers
Manuscript ID: polymers-2311460
Manuscript title: Identification of cement pavement with temperature effect and evaluation of polymer grouting effect
Author: Xifeng Du, Haoyuan Cheng, Shengjie Xu and Wenjun Pei
Corresponding author: Xifeng Du, 202012222014206@gs.zzu.edu.cn
Dear Reviewer,
Thanks for your comments on our article " Identification of cement pavement with temperature effect and evaluation of polymer grouting effect " (ID: polymers-2311460). These opinions greatly value our paper's revision and improvement and have important guiding significance for our research. We have carefully studied these comments and tried to revise and improve the manuscript. We would like your approval.
We numbered the reviewer's suggestions and comments in italics for ease of differentiation. The primary corrections and responses to the reviewer's comments are listed below. All the changes are included in the new manuscript. To compare traces of change, please upload a copy of the manuscript for reference.
Thank you again for your time and consideration. Looking forward to hearing from you.
Sincerely yours,
Xifeng Du
Corresponding author
Responds to the reviewer’s comments:
- Response to comment: It appears that there is no correlation between the bottom temperature and temperature difference across all seasons and these data were not used in any further analysis. What is the purpose of analyzing it?
Response: Thank you very much indeed for your comments. As cement concrete pavement slab is an imperfect thermal medium, the change of ambient temperature field around the slab has little influence on the temperature of the bottom of the slab. There is no apparent correlation between the temperature difference between the bottom of the cement concrete slab and the top of the slab, so we do not consider the correlation between the bottom of the slab and the temperature difference, which provides the corresponding theoretical basis for setting the temperature and grid density in the later simulation.(Lines 154-160)
- Response to comment: Figs 5, 7, 9, 11 plotted the temperature difference versus top temperature while description in line 135, 137, 139, 141 has no reference object. For example, we usually say ‘correlation between A and B is demonstrated…’ but description in line 135, 137, 139, 141 has the format as ‘correlation between A is…’. It appears as a grammar mistake but if I understand incorrectly, please correct me.
Response: Thank you very much indeed for your comments. Because our chart needs to be clearer, it causes your misunderstanding. Figures 5, 7, 9, and 11 here describe the temperature difference between the top of the cement concrete panel and the bottom of the roof. In lines 140, 142, 144, and 146 of the revised article, we have further described the correlation between the temperature difference between the top of the slab and the bottom of the roof of the cement concrete pavement in detail.
- Response to comment: Line 198-209 listed 3 assumptions for developing multi-layer model. The assumption # 3 indicates that there is perfect contact between two layers, while in reality, defects in contact exist universally. What would be the ultimate effect if we have imperfect contact? My understanding is that stress concentration will occur, and the model author developed will not be valid anymore. Can authors comment on that?
Response: Thank you very much indeed for your comments. In practical engineering, the permeable layer and viscous layer should be set up between pavement structure layers to make each pavement structure layer complete contact. In the simulation of this paper, the disease-free pavement is in perfect contact, and we set the place with the disease to empty (Figure 22).
- Response to comment: The mesh density that authors used was 0.2. Why do authors choose this value? What will happen if we choose mesh density that is higher or lower than 0.2?
Response: Thank you very much indeed for your comments. The higher the mesh density, the higher the calculation precision, the more complex the calculation, and the longer the calculation time. The lower the mesh density, the faster the calculation speed, and the corresponding accuracy will decrease. Therefore, we should select grids satisfying the calculation accuracy in engineering applications and distinguish the importance of different model parts. The calculation accuracy of critical components and critical nodes should be improved, so the refined grid can be selected, while the regions far away from constraints and loads or the details less affected by conditions and loads can be discretized by a rough grid appropriately. Devote limited resources and time to critical parts and nodes of the structure. Considering that the temperature change of the base is small, the mesh density required for simulation in this paper is 0.2. The mesh division density of cement concrete pavement board is more significant, with a thickness of 0.05, and the mesh density in the plate corner load area is 0.025.
- Response to comment: Legends from Fig25 to Fig45 are not well defined. For example, Figure 26 only shows data in one color but the legends indicated there are 16 colored data. Same thing happens to all rest of the figures. Authors should fix this.
Response: Thank you very much indeed for your comments. Because the influence of temperature difference on the deformation of the center point of the plate angular load under different unloading conditions in the same season is very similar in magnitude, some of the data of different colors shown in the figure will overlap, so only one color is shown.
- Response to comment: Illustrations of Equation 11 from line 316 to line 320 indicates that Ltemperature stress is the combination deformation while Ltemperature stress and load is the deformation under temperature stress only. The subscript does not align with the description. Authors should double check.
Response: Thank you very much indeed for your comments. Our expression error is corrected in the article (lines 324-327).
- Response to comment: Equations 13, 14,15,16 shows the correlation between influence coefficient and temperature difference. It very strange that authors displace R2value for these equations since the equations are clearly not demonstrating linear correlation. It is the basic statistical analysis knowledge that R2can only be used in linear correlation and the display of R2 here is scientifically wrong. Authors can do chi squared test or other statistical analysis instead.
Response: Thank you very much indeed for your comments. We realized the error of using and corrected it in the paper. We use RMSE and MAE to analyze the fit degree of the curve. (Lines 380-385 and 454-456).
- Response to comment: What is DelT0 and DelT in Equations 13, 14,15,16?
Response: Thank you very much indeed for your comments. is the temperature difference at the bottom of the panel is -1℃. DelT is the temperature difference. is the temperature difference at the bottom of the panel, and the top plate is 1℃. We have added the formula comments to lines 383-385.
Comments and Suggestions for Authors: This work illustrates the correlation between temperature and pavement deformation. There are many simulation data displayed while limited experimental data is displayed except for the temperature.
General comment: authors should consider including the experimental deformation data to demonstrate the feasibility of the model. Without the experimental data, any conclusion drawn from this study is not justified. In addition, there are many grammar mistakes and authors should consider seeking for professional editing help.
Response: Thank you very much indeed for your comments. We added Chapter 4.3, experimental verification, in this paper to prove the feasibility of the model. Simultaneously we have asked Dr. Bei Zhang, a well-established expert, to polish our paper. Please see if the revised version met the English presentation standard.

Round 2
Reviewer 3 Report
The authors have addressed all my previous comments and I have no more technical comments. However, minor grammar fix is still needed for this manuscript.